# ACTIONREASONINGBENCH: REASONING ABOUT ACTIONS WITH AND WITHOUT RAMIFICATION CONSTRAINTS

**Divij Handa**[*1]**, Pavel Dolin**[*1]**, Shrinidhi Kumbhar**[*1]**, Tran Cao Son**[2]**, Chitta Baral**[1]
[1]Arizona State University, [2]New Mexico State University
{dhanda,pdolin,skumbha4,chitta}@asu.edu, stran@nmsu.edu

## ABSTRACT

Reasoning about Actions and Change (RAC) has historically played a pivotal role in solving foundational AI problems, such as the frame problem. It has driven advancements in AI fields, such as non-monotonic and commonsense reasoning. RAC remains crucial for AI systems that operate in dynamic environments, engage in interactive scenarios, or rely on commonsense reasoning. Despite substantial advances made by Large Language Models (LLMs) in various AI domains, their performance in RAC remains underexplored. To address this gap, we introduce a new diagnostic benchmark, ACTIONREASONINGBENCH, which encompasses 8 domains and includes questions for up to 19 action sequences. This benchmark rigorously evaluates LLMs across six key RAC dimensions: *Fluent Tracking*, *State Tracking*, *Action Executability*, *Effects of Actions*, *Numerical RAC*, and *Composite Questions*. LLMs demonstrate average accuracy rates of 73.55%, 65.63%, 58.73%, and 62.38% on the former four dimensions, which are frequently discussed in RAC literature. However, the performance on the latter two dimensions, which introduce complex and novel reasoning questions, the average performance of LLMs is lowered to 33.16% and 51.19%, respectively, reflecting a 17.9% performance decline. We also introduce new ramification constraints to capture the indirect effects of actions, providing deeper insights into RAC challenges. Our evaluation of state-of-the-art LLMs, including both opensource and commercial models, reveals challenges across all RAC dimensions, particularly in handling ramifications, with GPT-4o failing to solve any question and o1-preview achieving a score of only 18.4%.

## 1 INTRODUCTION

Reasoning about actions and change (RAC) is a fundamental problem in artificial intelligence, with its roots tracing back to early work from the 1960s (McCarthy et al., 1963). Initially, research focused on developing logical systems capable of effectively modeling and reasoning about actions and their effects in a dynamic environment. One of the significant challenges in this domain has been succinctly expressing how actions influence changeable properties of the world, known as **fluents**. For example, consider the statement: "Moving an object from location X to location Y results in the object being at location Y." While it is relatively straightforward to describe the direct effects on the affected fluents, such as the object's location, it is much more complex to account for the unaffected fluents, a challenge known as the *frame problem*. This challenge becomes exacerbated when the descriptions involve relationships between fluents in a state, such as "an object can not be at two different places at the same time". While such constraints simplify action descriptions by decoupling them from fluents, they introduce indirect effects, or **ramifications**. For example, the statement "A block is said to be clear if there isn't any block on top of it" describes a ramification fluent, "clear" dependent on another fluent "on top of."

It took multiple decades of research to create a comprehensive logical formalization that adequately addressed these issues. It involved the laborious creation of numerous handcrafted rules and logic

---

*Equal contribution.

detailing the effects and preconditions of actions (Reiter, 2001). However, these tools are limited since they rely on manual effort to translate natural language descriptions of actions and their effects into formal logic representations. To address this challenge, recent research in natural language processing (NLP) has begun exploring the capabilities of LLMs in RAC tasks, as demonstrated by works of He et al. (2023), Spiliopoulou et al. (2022), and Banerjee et al. (2020). However, these studies have not systematically decomposed the complex RAC problem into multiple categories and overlook the critical ramifications of actions seen in real-world scenarios. To address this gap, we introduce ACTIONREASONINGBENCH, a diagnostic RAC benchmark that aims to pinpoint where modern state-of-the-art LLMs struggle.

We decompose the RAC task into six distinct categories—*Fluent Tracking*, *State Tracking*, *Action Executability*, *Effects of Actions*, *Numerical RAC*, and *Composite Questions*. The first four categories focus on assessing fundamental aspects of RAC, while the latter two introduce more complex and novel question types. The questions in every category span action sequences ranging from 1 to 19 steps, allowing us to test the RAC capabilities at a series of action sequence ranges. Additionally, we introduce ramification constraints to represent the indirect effect of actions. These constraints simplify action descriptions and align more closely with real-world conditions but introduce additional complexity, as highlighted by McIlraith (2000). Specifically, we expand the domains by adding ramification fluents with dependencies up to four levels deep, where actions propagate their effects through multiple layers.

Highlights of our benchmark, ACTIONREASONINGBENCH, along with the comparison to previous benchmarks on RAC, are presented in Table 1. We evaluate four LLMs–two open-source models, Llama-3.1-8B-Instruct and Llama-3.1-70B-Instruct (Dubey et al., 2024), as well as two leading proprietary models, GPT-4o (Achiam et al., 2023) and o1-preview (OpenAI, 2024). These LLMs were tested on ACTIONREASONINGBENCH across various RAC categories under different prompt settings, including Zero-shot-CoT (Kojima et al., 2022) and Few-shot-3 (Brown, 2020), to assess how performance varies based on these configurations.

Our findings indicate that LLMs face substantial challenges, particularly when addressing complex RAC questions. The average performance of all LLMs on the complex categories decreases by 17.88% compared to the first four basic categories. The best performing LLM, GPT-4o, achieves an average accuracy of 59.91% on these categories. Notably, GPT-4o failed to produce any correct answers for questions involving ramifications constraints, while the o1-preview model achieved an accuracy of only 18.42%. Performance was especially poor in categories like *Action Executability*, *Numerical RAC*, and *Composite Questions*, with further declines observed as the length of action sequences increased. Additionally, LLMs struggled with reasoning in scenarios that combined both true and false fluents, experiencing an average performance drop of 12.16% compared to tasks involving exclusively true or false fluents.

|  | **PlanBench** | **TRAC** | **(Ours)** |
|---|---|---|---|
| Number of domains | 2 | 1 | 8 |
| Number of queries | 26k | 15k | 152k |
| Max Action Sequence length | 48 | 3 | 19 |
| Max number of objects | 24 | 5 | 28 |
| Binary Questions (T/F) | ✗ | ✓ | ✓ |
| Free Answers | ✓ | ✗ | ✓ |
| State Tracking | ✓ | ✓ | ✓ |
| Action Executability | ✓ | ✓ | ✓ |
| Fluent Tracking | ✗ | ✗ | ✓ |
| Effects of Actions | ✗ | ✗ | ✓ |
| Numerical Reasoning | ✗ | ✗ | ✓ |
| Composite Questions | ✗ | ✗ | ✓ |
| Ramifications Constraints | ✗ | ✗ | ✓ |
| Subcategories of Fluents | ✗ | ✗ | ✓ |

Table 1: Differences between ACTIONREASONINGBENCH (Ours) and previous benchmarks on RAC. PlanBench (Valmeekam et al., 2024) ; TRAC (He et al., 2023)

## 2  RELATED WORKS

**Benchmarking reasoning capabilities of LLMs**    Evaluating the reasoning capabilities of LLMs using synthetic datasets has become a key focus in NLP, with increasing efforts to create challenging benchmarks. Notable areas of interest include the legal reasoning (Fei et al., 2023; Guha et al., 2023), logical reasoning (Luo et al., 2024; Han et al., 2024; Patel et al., 2024; Parmar et al., 2024) arithmetic

reasoning (Cobbe et al., 2021; Miao et al., 2021), temporal reasoning (Uddin et al., 2024; Fatemi et al., 2024), and commonsense reasoning (Onoe et al., 2021; Lin et al., 2021; Geva et al., 2021; Lourie et al., 2021). Despite this progress, RAC remains significantly under-explored, even though it plays a crucial role in several of these reasoning tasks, such as commonsense and legal reasoning. To fill this gap, we create ACTIONREASONINGBENCH using synthetically generated data.

**Evaluating RAC and Planning**   Benchmarking planning capabilities of LLMs is a well-studied area, with recent works (Zheng et al., 2024; Xie et al., 2024) demonstrating the challenges LLMs face. While planning is a non-polynomial problem that remains inherently difficult to solve, we believe that RAC, which is a polynomial problem, is a prerequisite for effective planning. Without comprehending the effects of actions, LLMs are unlikely to construct coherent plans. In this work, we address this gap by proposing ACTIONREASONINGBENCH, a diagnostic benchmark for RAC.

Previous research, such as Banerjee et al. (2020), has investigated RAC capabilities in models like RoBERTa, focusing primarily on binary questions or single-word answers. Extending this, He et al. (2023) assessed LLMs on a broader range of question categories. However, only two of these—Action Executability and State Tracking—pertain directly to RAC, with the remainder addressing the broader domain of planning. Similarly, Valmeekam et al. (2024) introduced a benchmark evaluating both RAC and planning, with a detailed emphasis on planning tasks but maintaining a limited focus on RAC categories, specifically State Tracking and Action Executability. A comparative analysis of the benchmarks from He et al. (2023), Valmeekam et al. (2024), and our proposed benchmark is summarized in Table 1.

Parallel to our work, Kokel et al. (2024) introduced ACPBench to evaluate both RAC and planning. While ACPBench overlaps with our work, it primarily covers fundamental aspects of RAC. In contrast, our benchmark introduces greater complexity by incorporating categories such as State Tracking, Numerical RAC, Composite Questions, and constraints related to ramifications.

In this work, we concentrate on RAC and develop a benchmark that encompasses a wider range of categories, enabling more precise identification of areas where LLMs underperform. Furthermore, our benchmark introduces novel constraints involving ramifications, incorporating the indirect effects of actions. To the best of our knowledge, ramification constraints have not been addressed in any existing benchmark, marking a significant advancement in the evaluation of RAC capabilities.

## 3   ACTIONREASONINGBENCH

This section provides a detailed overview of our benchmark, including its categorization, creation methodology, and validation process. A sample instance is presented in Appendix B, where we also describe the objects, actions, and fluents within the domain.

### 3.1   QUESTION CATEGORIES

In our benchmark, the questions are organized into six distinct categories, each designed to assess a specific dimension of RAC. Below, we provide a detailed description of each category.

1. **Fluent Tracking** - Given the initial state and the sequence of actions performed, this category contains questions about the fluents, i.e. properties of the domain, of an object from the changed state. For instance, in the *Grippers* domain, a fluent-tracking question might be *"List all valid properties associated with ball2."*

2. **State Tracking** - This category extends the concept of *Fluent Tracking*. It involves querying about the complete set of fluents in the final state. For instance, in the *Blocksworld* domain, a state-tracking question might be *"What are all the valid properties in this state?"*

3. **Action Executability** - This category encompasses two types of questions related to executability of actions. The types of questions within this category are as follows:

   (a) Given an initial state, and a sequence of actions, the question focuses on identifying the first action in the sequence that is not executable.

   (b) Given an initial state and a sequence of actions leading to a final state, the task is to identify all actions that can be executed in the final state. For instance, in the *Visitall*

domain, an action-executability question might be *"List all executable actions present in the current state."*

4. **Effects of Actions** - This category contains questions that explore the outcomes of performing a specific action. For instance, in the *Mystery* domain, an Effects-of-action question can be *"From the current state, the vehicle v0 moves from location l1 to l0, and has fuel-level f6 and f5, which properties of the state will be true now?"*

5. **Numerical RAC** - Questions requiring a numerical response fall under this category. These questions may derive from any of the four previously mentioned categories. For example, in the *Spanner* domain, a Numerical-RAC question can be *"What are the number of executable actions in the current state?"*

6. **Composite Question** - This category contains questions that integrate multiple above-mentioned categories, combining up to three distinct categories. These questions require multiple steps of reasoning to arrive at the correct solution. For example, in the *Satellite* domain, a composite question may combine aspects of *Fluent Tracking* and *Action Executability*. An example of such a question could be *"List all the properties of the state for satellite0 before the first infeasible action in the sequence?"*

## 3.2 FLUENT CATEGORIES

We further divide the fluents of all 8 domains into three distinct categories, each representing a different aspect of ramifications within RAC.

1. **Static Properties** - These properties remain unchanged regardless of any action performed. For instance, the property *"connected"* in the *Visitall* domain represents whether two locations are connected, a relationship that remains constant irrespective of the robot's action, which may involve moving, picking up, or placing down objects. In this domain, the connectivity between locations remains unchanged, irrespective of any action.

2. **Base Fluents** - These fluents can change as a direct result of an action and do not depend on other fluents. For example, in the *Grippers* domain, the fluent *"carry"* indicates whether an object is being carried by a robot's gripper. This fluent can change if the action pick or drop is performed.

3. **Ramification Fluents** - These fluents are influenced indirectly by other fluents, and action descriptions do not explicitly dictate their changes. Instead, they are determined by the dependencies and relationships within the system. Ramification fluents are further divided into two sub-categories:

    (a) **Derived Fluents** - These fluents rely on the state of other fluents, reflecting a level of dependency. Changes to them occur as a consequence of changes in the fluents they depend on rather than through direct action. For instance, the fluent *"stable"* in the *Blocksworld* domain is considered a derived fluent as its state depends on the fluents *"clear"* and *"on_table"*. This relationship can be described as: *"Blocks are stable when clear and on the table"*. Furthermore, fluent *"clear"* is itself a derived fluent, dependent on the fluent *"on"*, which makes *"stable"* a second-level indirect effect.

    (b) **Self-Derived Fluents** - These fluents rely on constraints related to themselves rather than other fluents. For example, in the *Depot* domain, the fluent *"at"* represents the location of a truck, which can only be at one location at any given time. If the truck is *at* location l0, it cannot simultaneously be *at* location l1. Such constraints are explicitly included in the domain description, for example, *"A truck can only be in one location at a time"*.

Classification of every fluent across all 8 domains can be found in Appendix I. Furthermore, for each fluent type, we generate questions involving negative fluents, i.e. fluents that are false, which allows us to evaluate LLM's comprehension of negation within RAC contexts.

## 3.3 DATASET STRUCTURE AND VARIATIONS

**Selected Domains** ACTIONREASONINGBENCH requires domains that facilitate the evaluation of LLMs on both short and long sequences of meaningful actions, where the effects and preconditions

of these actions are succinctly described. Additionally, these domains should reflect real-world scenarios. To meet these criteria, we selected 8 domains *Blocksworld*, *Depots*, *Driverlog*, *Grippers*, *Mystery*, *Satellite*, *Spanner*, and *Visitall*–sourced from the International Planning Competition (IPC), covering the years 1998 to 2014. These domains are commonly used as benchmarks for evaluating advanced planning systems and provide a robust foundation for research in automated planning. Appendix G provides a detailed description of each domain. Notably, even state-of-the-art LLMs like GPT-4o are not capable of generating diverse domains or action sequences that conform to the precise constraints outlined in these domain descriptions, justifying the reliance on IPC domains.

**Domain Descriptions and Ramifications**    The domains provided by the IPC are described using the Planning Domain Definition Language (PDDL), a formal language designed to model deterministic actions and state transitions for planning problems. Further information on PDDL can be found in Appendix H.1. In this study, we initially translated the PDDL-based domains into natural language. Subsequently, ramification constraints were introduced into these natural language descriptions. The process underwent validation by two experts in the RAC domain to ensure correctness. Given that the category *Action Executability* focuses on determining whether the action can be performed rather than analyzing its effect, we concentrated on the categories *Fluent Tracking*, *State Tracking*, and *Effects of Actions* when introducing ramifications.

**Action-Sequence Lengths**    In order to fine-tune LLMs, we generate a comprehensive set of questions that span various action-sequence lengths, specifically 1, 5, 10, 15, and 19. This range is chosen to capture increasing complexities in RAC. We curate a distinct subset of questions with action-sequence lengths of 1, 10, and 19 for evaluation. This subset is selected to assess the model's performance across the action sequences range.

**Answer Types**    We formulate two distinct types of questions based on the expected answer format. The first type consists of binary questions, where the response is either True or False. The second type involves subjective answers, which encompass a range of multiple objects, actions, or fluents.

## 3.4    DATA CREATION & VALIDATION

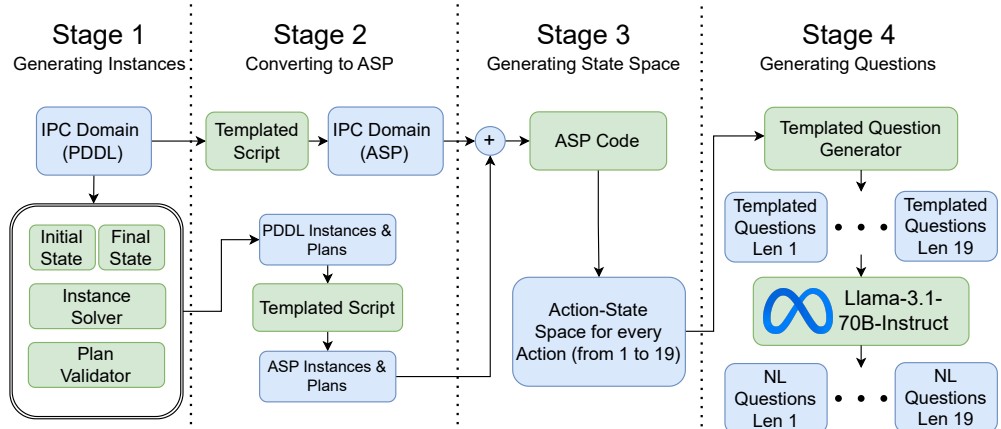

Figure 1: Overview of the question generation pipeline for ACTIONREASONINGBENCH. Blue blocks represent "Generated Data", and green blocks represent "Code used in the pipeline". Stage 1 involves generating states and plans using Helmert (2006) and validating them with Howey et al. (2004). In Stage 2, PDDL instances and plans are converted to ASP. Stage 3 computes the action-state space through ASP. Stage 4 generates questions using templates, which are then rephrased to natural language via Llama-3.1-70B-Instruct.

The question generation process follows a four-stage pipeline, as illustrated in Figure 1. The selected domains from the IPC are represented in PDDL (see Appendix G for examples). First, these PDDL representations are used to generate 10 pairs of initial and goal conditions. A PDDL solver

(Helmert, 2006) and validator (Howey et al., 2004) are then employed to obtain and validate the action sequences necessary to transition from the initial to the goal state. In the second stage, the PDDL domains, instances, and action sequences are converted into Answer Set Programming (ASP) descriptions using Python-based templates.

In the third stage, ASP solvers are used to generate the action-state space and extract fluents for each state, along with identifying all executable and inexecutable actions. Further details on these formal languages are provided in Appendix H. Finally, the fourth stage involves converting the action-state data into questions using a Python template. Up to three natural language variations are created for every object, action, and fluent. These templated sequences are then paraphrased to Llama-3.1-70B-Instruct to ensure they sound natural and avoid repetition in long action sequences. Three independent annotators review both the templated and the paraphrased versions to assess their naturalness, as detailed in Appendix D. Additionally, all eight domain descriptions were manually translated from PDDL to natural language.

## 3.5 DATA SPLITS

The benchmark was divided into two parts: one for training and the other for testing the LLMs, ensuring a balanced representation of question categories across the 8 domains. The *Composite Questions* category is slightly larger in the test set, as it combines multiple categories, leading to increased questions. Table 2 provides an overview of the distribution of questions and their corresponding categories across both the training and testing sets. The test set contains 3,498 questions, including 2,195 binary and 1,303 free-answer questions. Finally, we designed both zero-shot-CoT and few-shot-3 prompts for all the questions in the test set.

|  | Test Set | Train Set |
|---|---|---|
| Fluent Tracking | 438 | 57,906 |
| State Tracking | 382 | 12,636 |
| Action Executability | 450 | 9,562 |
| Effects of Actions | 417 | 8,939 |
| Numerical Reasoning | 414 | 31,506 |
| Composite Questions | 1,397 | 28,688 |
| Static Properties | 237 | 12,458 |
| Base Fluents | 231 | 10,461 |
| Derived Fluents | 366 | 15,946 |
| Self-Derived Fluents | 390 | 23,436 |
| Mixed Fluents | 2,274 | 86,936 |
| Total Unique Questions | 3,498 | 149,237 |

Table 2: Overview of the test and train sets across Question and Fluent Categories. The "Mixed Fluents" category represents questions that involve more than one type of fluent.

## 4 EXPERIMENTS AND EVALUATION

**Models** To evaluate our benchmark, we tested four LLMs and employed two prompting techniques. The LLMs include two proprietary models–GPT-4o (Achiam et al., 2023) and o1-preview (OpenAI, 2024)–alongside two open-source models, Llama-3.1-8B-Instruct and Llama-3.1-70B-Instruct (Dubey et al., 2024). Each LLM was evaluated using both few-shot prompting with three examples (few-shot-3) Brown (2020) and zero-shot-CoT Kojima et al. (2022) prompting.

While the entire dataset requires reasoning abilities, the ramification subset involves the most complex and challenging reasoning tasks. Given that o1-preview is specifically optimized for reasoning tasks and incurs significantly higher costs compared to GPT-4o[1], we restricted its use to the ramification subset, where its superior reasoning capabilities are expected to provide the greatest benefit. Utilizing o1-preview across the entire dataset would not be cost-effective, as its advantages would be less pronounced for simpler reasoning tasks.

To ensure a fair evaluation of the standalone reasoning capabilities of LLMs, we deliberately avoided incorporating external tools or systems. Although integrating formal solvers (e.g., PDDL-based planners) with LLMs could potentially improve performance, our primary focus is on assessing the intrinsic reasoning abilities of these models.

**Evaluation & Metrics** ACTIONREASONINGBENCH includes two types of answer formats, as outlined in Section 3.3: binary (true/false) and free-form responses. The evaluation process was performed separately for each answer type. For binary questions, we extracted "true" and "false" keywords from the model's response and compared them to the ground truth. Since free-form answers can't be evaluated using exact string matching, we employed human evaluation for the ramification

---

[1]As of Oct 2024, o1-preview is six times more expensive than GPT-4o https://openai.com/api/pricing/

questions. While human evaluation is highly accurate, it is not scalable, so we used Llama-3.1-70B-Instruct to assess all free-form responses. The specific prompt used to evaluate the LLMs and the correlation between Llama-3.1-70B-Instruct and human evaluations are detailed in Appendix F . For all experiments, we report the accuracy along with the standard error of the mean (SEM), calculated as $SEM = \frac{\sigma}{\sqrt{n}}$, where $\sigma$ represents the standard deviation, and $n$ is the sample size.

**Fine-tuning** We fine-tuned the Llama-3.1-8B model using the training data split outlined in Section 3.5. Due to the limited computing power, we excluded any data samples that exceeded a context length of 4096 tokens. The fine-tuning process was performed separately for free-response and binary questions. Detailed information on the fine-tuning procedure is provided in Appendix E. All experiments were executed using 8×H100 GPUs.

## 5 RESULTS AND DISCUSSION

This section presents the results and analysis using ACTIONREASONINGBENCH. The Zero-shot-CoT results for each LLM on both the binary and free-response subsets of the test set are provided in Tables 5 and 3 respectively. Similarly, the Few-shot-3 results for each LLM evaluated on these same subsets are displayed in Tables 6 and 7. The detailed analysis of the effects when ramification constraints are incorporated into the descriptions is discussed in Section 5.1.

| Action Seq. | Ques Categories | GPT-4o | Llama-8B-Inst | Llama-70B-Inst | Finetuned Llama-8B |
|---|---|---|---|---|---|
| 1 | Fluent Tracking | $88.46_{4.43}$ | $30.77_{6.40}$ | $71.15_{6.28}$ | $76.92_{5.84}$ |
| | State Tracking | $73.33_{6.59}$ | $28.89_{6.76}$ | $64.44_{7.14}$ | $75.56_{6.41}$ |
| | Action Executability | $27.08_{6.41}$ | $08.33_{3.99}$ | $33.33_{6.80}$ | $31.25_{6.69}$ |
| | Effects of Actions | $82.50_{6.01}$ | $20.00_{6.32}$ | $67.50_{7.41}$ | $60.53_{7.93}$ |
| | Numerical RAC | $11.11_{4.68}$ | $06.67_{3.72}$ | $04.44_{3.07}$ | $08.89_{4.24}$ |
| | Composite Questions | $64.53_{3.36}$ | $24.63_{3.02}$ | $43.35_{3.48}$ | $72.41_{3.14}$ |
| | Average | $60.28_{2.35}$ | $21.71_{1.98}$ | $45.96_{2.39}$ | $61.02_{2.35}$ |
| 10 | Fluent Tracking | $82.00_{5.43}$ | $36.00_{6.79}$ | $62.00_{6.86}$ | $80.00_{5.66}$ |
| | State Tracking | $74.42_{6.65}$ | $18.60_{5.93}$ | $60.47_{7.46}$ | $66.67_{7.27}$ |
| | Action Executability | $34.09_{7.15}$ | $11.36_{4.78}$ | $40.91_{7.41}$ | $47.73_{7.53}$ |
| | Effects of Actions | $76.09_{6.29}$ | $19.57_{5.85}$ | $65.22_{7.02}$ | $62.22_{7.23}$ |
| | Numerical RAC | $10.20_{4.32}$ | $02.04_{2.02}$ | $06.12_{3.42}$ | $10.20_{4.32}$ |
| | Composite Questions | $59.11_{3.45}$ | $16.26_{2.59}$ | $45.32_{3.49}$ | $58.62_{3.46}$ |
| | Average | $57.01_{2.37}$ | $17.01_{1.80}$ | $45.98_{2.39}$ | $55.66_{2.39}$ |
| 19 | Fluent Tracking | $67.44_{7.15}$ | $27.91_{6.84}$ | $67.44_{7.15}$ | $67.44_{7.15}$ |
| | State Tracking | $75.51_{6.14}$ | $16.33_{5.28}$ | $51.02_{7.14}$ | $65.31_{6.80}$ |
| | Action Executability | $41.67_{7.12}$ | $08.33_{3.99}$ | $29.79_{6.67}$ | $37.50_{6.99}$ |
| | Effects of Actions | $76.60_{6.18}$ | $14.89_{5.19}$ | $46.81_{7.28}$ | $61.70_{7.09}$ |
| | Numerical RAC | $10.20_{4.32}$ | $06.12_{3.42}$ | $08.16_{3.91}$ | $06.12_{3.42}$ |
| | Composite Questions | $60.30_{3.47}$ | $08.54_{1.98}$ | $38.19_{3.44}$ | $49.25_{3.54}$ |
| | Average | $56.78_{2.38}$ | $11.72_{1.54}$ | $39.17_{2.34}$ | $48.05_{2.40}$ |

Table 3: Performance comparison of GPT-4o, Llama-3.1-8B-Instruct, Llama-3.1-70B-Instruct, and fine-tuned Llama-3.1-8B on the free-answer subset of the benchmark, evaluated without the ramifications constraints using the zero-shot-CoT. The results are categorized up by the action-sequence lengths and question categories.

**Performance across Domains** In our evaluation, GPT-4o demonstrated the highest performance on the *Grippers* domain and the lowest on the *Satellite* domain, with a performance gap of 15.53%. For both Llama-3.1-8B-Instruct and Llama-3.1-70B-Instruct, the best performance is also on the *Grippers* domain, but their worst performance occurs on the *Mystery* domain, with differences of 19.24% and 24.04%, respectively. Interestingly, the second lowest performing domain for GPT-4o is *Mystery*, while for the Llama models, it is *Satellite*. A detailed breakdown by domains is presented in Appendix C.3. This suggests that despite potential differences in pre-training data, these models exhibit a similar relative understanding of the domains.

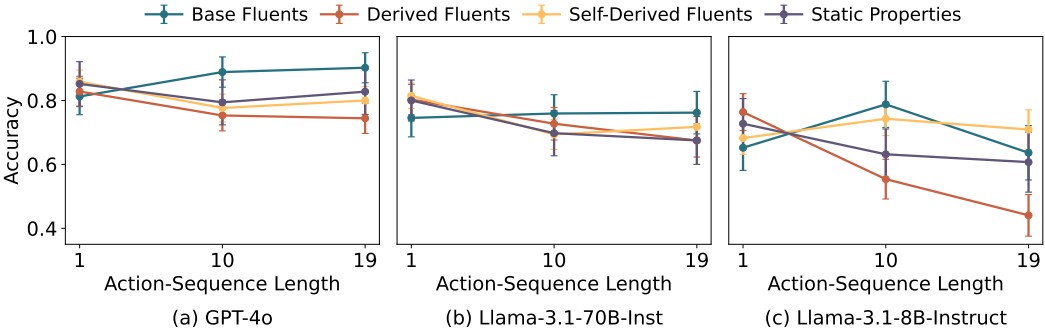

Figure 2: Performance for every Fluent Category for both binary and free-answer questions for every Action-Sequence length for GPT-4o, Llama-3.1-70B-Instruct, and Llama-3.1-8B-Instruct for Zero-shot-CoT prompt. Note: Bars represent SEM.

Furthermore, when correlating these results with the State Space Complexity metric (described in App. G), traditionally used to gauge complexity for classical AI systems, we observe an inconsistent trend. This discrepancy implies that LLMs may employ different heuristics from those of traditional AI systems when tackling RAC problems, an observation that opens avenues for future exploration.

**Performance across Question Categories**  LLMs perform well in *Fluent Tracking*, *State Tracking*, and *Effects of Actions*, demonstrating their strength in keeping track of changes. However, they struggle with *Action Executability*, *Composite Questions*, and *Numerical RAC*. The performance of all LLMs on free-response complex questions highlights significant challenges, especially in the *Numerical RAC* category. This category reformulates existing question types into numerical formats, a domain where all tested LLMs exhibit notable difficulty. Specifically, performance on numerical questions related to *Action Executability* is 8.65% lower than on questions in the *Fluent Tracking* category. Previous research, such as (Ahn et al., 2024; McCoy et al., 2023), indicates that LLMs struggle with arithmetic reasoning and counting, which, when mixed with the RAC questions, likely contributes to the poor performance in the *Numerical RAC* category.

For *Composite Questions*, the combination of *Fluent Tracking* and *Action Executability* proves easier to answer than the combination of *State Tracking* and *Action Executability*, with a 16.32% performance difference. This can be attributed to the fact that the *State Tracking* category is a superset of the *Fluent Tracking* category, thereby explaining the observed difference in difficulty.

**Performance across Fluent Categories**  As evident from Figure 2 and Table 9, across all LLMs examined in the study, a consistent trend emerges in which performance on *Static Properties* decreases as the length of action-sequence increases. While these static properties remain unchanged throughout the actions, they might get overlooked in longer sequences, likely due to their absence in the effect of any action. This phenomenon resembles the "needle in a haystack" challenge in long-context scenarios, where LLMs struggle to recall specific information embedded within a long context (Zhang et al., 2024). Conversely, *Base Fluents* maintain stable performance across all action sequences, indicating that the LLMs consistently capture the direct effects of actions. *Ramification fluents* exhibit a steady decline in performance as the sequence lengthens, particularly affecting the subcategory of *Derived Fluents*, which suggests that LLMs have more difficulty handling indirect effects. Finally, *Mixed Fluents*, which involve more than one fluent type, show a consistent decline in performance as the length of action sequences increases.

**Performance across Action-Sequence Lengths**  Figure 3 illustrates the performance of the three models—GPT-4o, Llama-3.1-70B-Instruct, and Llama-3.1-8B-Instruct—across varying action sequence lengths in a zero-shot-CoT setting. The results combine both binary and free-response formats, with detailed data available in Tables 5 and 3. Generally, model accuracy declines as the action-sequence length increases, a pattern that holds for most categories. However, GPT-4o deviates from this trend in the *State Tracking* category, where performance first decreases and then improves. Since this trend is not observed with the other two models, and the results lie within the

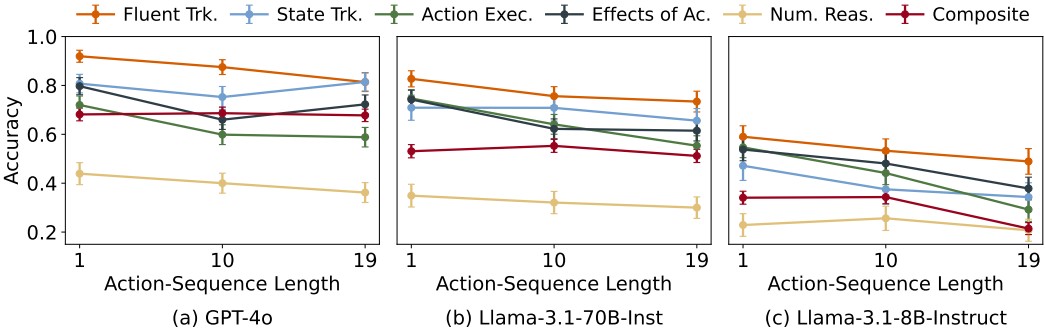

Figure 3: Performance for every QUestion Category for both binary and free-answer questions for every Action-Sequence length for GPT-4o, Llama-3.1-70B-Instruct, and Llama-3.1-8B-Instruct for Zero-shot-CoT prompt. Note: Bars represent SEM.

margin of error, we believe this is an outlier. In contrast, the *Effects of Actions* category consistently deviates from this trend, likely due to the nature of the task, which focuses on changes resulting from the last action, making it less dependent on the sequence of actions.

**Model Parameters and Fine-tuning**   As demonstrated in Tables 5 and 3, the Llama-3.1-70B-Instruct model consistently outperforms the smaller Llama-3.1-8B-Instruct model, with an average performance improvement of 20.84%. This improvement is likely due to the larger model's superior reasoning capabilities stemming from its increased number of parameters. A similar trend is observed when comparing GPT-4o to Llama-3.1-70B-Instruct, where GPT-4o exhibits an average performance increase of 8.64%. Although the specific size of GPT-4o remains undisclosed, it is widely speculated to be in the trillions of parameters[2]. Notably, fine-tuning the Llama-3.1-8B model on the training set resulted in substantial gains in both binary and free-answer tasks, with an average performance increase of 33.68% across the test set, even outperforming GPT-4o by 4.2%.

**Impact of Few-Shot Examples on Model Performance**   As shown in Tables 5 and 6, the inclusion of few-shot examples for binary answer categories does not significantly enhance model accuracy. This limitation is especially pronounced in models such as GPT-4o and Llama-3.1-70B-Instruct, which exhibit a relative performance decline of approximately 3.5% compared to zero-shot-CoT conditions. We hypothesize that this decrease may be attributed to the few-shot examples inadvertently leading the model toward detecting spurious correlations. In contrast, the model relies more heavily on its internal reasoning capabilities in a zero-shot-CoT setting, potentially mitigating biases introduced by example-driven patterns. However, as seen from Tables 3 and 7, the few-shot approach shows effectiveness in free-answer questions only open-source LLMs.

**LLMs Struggle with Negative Fluents**   Our study reveals a consistent pattern across all the LLMs examined as demonstrated in Table 10, wherein their performance declines when tasked with questions involving negative fluents compared to those focused on fluents that are true. Specifically, we observed a 12.16% decrease in accuracy on these negative fluent tasks. Furthermore, when questions required reasoning about both true and false fluents simultaneously, LLMs exhibited competence in identifying the true fluents but demonstrated difficulty in correctly recalling the false ones.

### 5.1   RAMIFICATIONS RESULTS

As discussed in Section 4, the performance on ramification fluents is evaluated for two LLMs: GPT-4o, the highest-performing LLM, and o1-preview, the most recent state-of-the-art LLM. Table 4 presents the performance of both models when ramification constraints are introduced. Further examples of the model responses to ramification-related questions can be found in the Appendix J.

---

[2]GPT-4o and Gemini 1.5 Pro: How the New AI Models Compare - CNET

| Action Sequence | Question Categories | Free Answer | | Binary Questions | |
|---|---|---|---|---|---|
| | | GPT-4o | o1-preview | GPT-4o | o1-preview |
| 1 | Fluent Tracking | $00.00_{00.00}$ | $00.00_{00.00}$ | $71.43_{17.10}$ | $100.00_{00.00}$ |
| | State Tracking | $00.00_{00.00}$ | $20.00_{17.89}$ | $100.00_{00.00}$ | $100.00_{00.00}$ |
| | Effects of Actions | $00.00_{00.00}$ | $40.00_{21.91}$ | $71.43_{17.10}$ | $57.14_{18.70}$ |
| | Average | $00.00_{00.00}$ | $25.00_{12.49}$ | $80.95_{08.57}$ | $85.71_{07.60}$ |
| 10 | Fluent Tracking | $00.00_{00.00}$ | $00.00_{00.00}$ | $57.14_{18.70}$ | $57.14_{18.70}$ |
| | State Tracking | $00.00_{00.00}$ | $33.33_{19.24}$ | $42.86_{18.70}$ | - |
| | Effects of Actions | $00.00_{00.00}$ | $14.28_{13.22}$ | $71.43_{17.10}$ | $100.00_{00.00}$ |
| | Average | $00.00_{00.00}$ | $23.07_{11.68}$ | $57.14_{10.80}$ | $78.57_{10.90}$ |
| 19 | Fluent Tracking | $00.00_{00.00}$ | $33.33_{27.21}$ | $42.86_{18.70}$ | $57.14_{18.70}$ |
| | State Tracking | $00.00_{00.00}$ | $00.00_{00.00}$ | $57.14_{18.70}$ | - |
| | Effects of Actions | $00.00_{00.00}$ | $00.00_{00.00}$ | $71.43_{17.10}$ | $85.71_{13.20}$ |
| | Average | $00.00_{00.00}$ | $07.69_{07.38}$ | $57.14_{10.80}$ | $71.43_{12.10}$ |

Table 4: Performance comparison of GPT-4o and o1-preview on both the free-answer and binary question subset of the benchmark, evaluated with the ramifications constraints using the zero-shot-CoT. The results are categorized up by the action-sequence lengths and question categories with a "-" indicating no response (due to longer prompts). We have provided some responses in App. J.

**GPT-4o Performance** GPT-4o did not answer any ramification-related questions correctly, as depicted in Table 4. Upon manual inspection of its outputs, it became evident that GPT-4o frequently failed to mention ramification fluents, even when these were explicitly detailed in the domain description. In instances where it did address ramification fluents, the responses were incorrect or incomplete, with some fluents being omitted. We hypothesize that GPT-4o may have encountered the domain data during pre-training and relied on memorized effects of actions, as the experimental domains were derived from publicly available IPC datasets. Since the ramification fluents were manually created and integrated for this study, this evaluation assesses the model's reasoning abilities without leveraging pre-existing knowledge. This likely explains why GPT-4o failed to generate ramification fluents, as its pre-training included only the original fluents from the domains.

**o1-preview Performance** o1-preview, a recently developed LLM optimized for reasoning tasks and incorporating a novel run-time inference mechanism OpenAI (2024), demonstrated significantly better performance on ramification-related questions compared to GPT-4o, as presented in Table 4. A detailed review of its outputs showed that o1-preview can correctly identify most ramification fluents. However, the model struggles with fluents involving negation, which consistently poses a challenge. It often omitted certain fluents in its final answers and, in some cases, incorrectly evaluated the ramification fluents.

## 6 CONCLUSION

In this work, we introduced a new diagnostic benchmark, ACTIONREASONINGBENCH, designed to comprehensively evaluate the performance of large language models (LLMs) on reasoning about actions and change (RAC). By assessing various LLMs across eight domains and six key dimensions of RAC, our findings indicate that while LLMs demonstrate moderate proficiency on traditional RAC tasks, such as *Fluent Tracking*, *State Tracking*, *Action Executability*, and *Effects of Actions*, they exhibit significant challenges when addressing more complex and novel questions, particularly in areas like *Numerical RAC* and *Composite Questions*, with an average performance drop of 17.9%. This highlights a substantial gap in the current ability of LLMs to handle complex reasoning tasks.

Additionally, we explored the indirect effects of actions, known as ramifications, where even state-of-the-art models show considerable limitations. For example, GPT-4o could not solve any questions involving ramifications, and the o1-preview model achieved a low score of 18.4%. These results underscore the pressing need for further research and advancements in RAC reasoning, especially in addressing indirect effects and more advanced reasoning tasks.

ACKNOWLEDGEMENT

We thank the anonymous reviewers for their constructive suggestions.

We extend our gratitude to the Research Computing (RC), and Enterprise Technology at ASU for providing computing resources, and access to the ChatGPT enterprise version for experiments.

We acknowledge support by a 2023 Spring Amazon Research Award (ARA). This material is also based upon work supported by the Engineering Research and Development Center - Information Technology Laboratory (ERDC-ITL) under Contract No. W912HZ24C0022.

Son Tran acknowledges the partial support from NSF grant #1914635 during the time working on this paper.

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

## A    LIMITATIONS & FUTURE WORK

While ACTIONREASONINGBENCH provides a diagnostic assessment of LLMs on RAC, it represents an early step in this area and has several limitations, including but not limited to the following:

1. Although RAC is not inherently dependent on the English language, the current version of ACTIONREASONINGBENCH is limited to questions formulated in English.

2. Although there are more complex types of RAC, including incorporating more reasoning types, exploring those remains beyond the scope of this work and is left as a direction for future research.

3. While the IPC domains in our work cover many scenarios, they may introduce a bias towards planning-specific domains. Expanding the dataset to include more domains could help mitigate this bias.

4. Despite our efforts to evaluate a variety of LLMs, including open and proprietary LLMs, our assessment did not cover models with different architectures or training approaches due to resource limitations.

5. Recent studies by Long et al. (2024) and Tam et al. (2024) indicate that LLM performance can fluctuate depending on the prompt format. This variation may lead to a marginal improvement in the performance of LLMs on ACTIONREASONINGBENCH.

6. Our free-answer evaluation, which relies on prompting Llama-3.1-70B-Instruct, isn't perfect and reflects an ongoing challenge of evaluating the free-answers within the NLP community.

## B    DESCRIBING AN INSTANCE FROM ACTIONREASONINGBENCH

---

**Blocksworld domain with Ramifications for a single sequence of action for Fluent Tracking**

[DOMAIN DESCRIPTION]
A block can only be picked up if it is clear, on the table, and the hand is empty, resulting in the block being held. A held block can be put down, placing it back on the table. Blocks can be stacked if the first block is held and the second block is clear, causing the first block to rest on top of the second. Unstacking occurs when the hand is empty, the first block is clear, and on top of the second, resulting in the first block being held again. A block can't be at two locations at the same time and is considered clear if nothing is on top of it and it's not held, and the hand is empty if it's not holding anything. Blocks are stable when clear and on the table, and they can be painted if stable and the hand is empty. A block is considered on display if it can be painted and has no other block on top of it.

[INITIAL CONDITIONS]
Block b1 is situated on the table, block b2 is not stacked with any other block, block b2 is also on the table, block b3 is not stacked with any other block, block b3 is positioned on top of block b7, block b4 is stacked on top of block b1, block b5 is not stacked with any other block, block b5 is placed on top of block b4, block b6 is on the table, block b7 is stacked on top of block b6, and the hand is empty.

[QUESTION]
Starting from the initial condition, the following actions are taken: block b3 is unstacked from the top of block b7 to achieve the current state. In this state, what are the valid properties (including both affirmative and negated properties) for b7? If there are no valid properties, write None.

---

In the domain description, actions and their corresponding effects on the state are outlined, including the necessary conditions for performing these actions. The initial conditions describe the starting configuration of objects within the domain. A typical scenario involves executing a sequence of actions that alter the configuration of the state, followed by a query. In the example provided, the

question falls under the category of fluent tracking, which asks about a specific set of properties associated with the object "block b7" after one action has been performed.

The objects involved in this example are as follows:

- block b1
- block b2
- block b3
- block b4
- block b5
- block b6
- block b7
- hand

The properties of the "block" object include:

- Block X on top of block Y
- Block X is on the table
- Block X is clear
- Block X is stable
- Block X can be painted
- Block X can be displayed
- Block X is held

The properties of the "hand" object are:

- The hand is empty
- The hand is holding block x

Several ramification constraints (i.e., properties that depend on other properties) are present in "Blocksworld":

- **Clear**: depends on the properties "on top of" and "held"
- **Stable**: depends on the properties "clear" and "on the table"
- **Paint**: depends on the properties "stable" and "hand is empty"
- **Display**: depends on the properties "painted" and "on top"
- **On top of**: depends on itself, since a block can't be at two locations at the same time
- **Hand is holding a block**: depends on itself, since the hand cannot hold two blocks at the same time

The valid actions that can be performed within this domain include:

- Picking up a block
- Putting down a block
- Stacking a block on top of another block
- Unstacking a block from top of another block

## C    ADDITIONAL RESULTS

### C.1    ZERO-SHOT-COT BINARY RESULT

Table 5 shows the results on GPT-4o, Llama-3.1-8B-Instruct, and Llama-3.1-70B-Instruct on zero-shot-CoT prompting.

| Action Seq. | Ques Categories | GPT-4o | Llama-8B-Inst | Llama-70B-Inst | Fine-tuned Llama-8B |
|---|---|---|---|---|---|
| 1 | Fluent Tracking | $94.44_{2.70}$ | $80.00_{4.78}$ | $90.12_{3.31}$ | $89.06_{3.90}$ |
|  | State Tracking | $85.94_{4.35}$ | $80.00_{8.00}$ | $79.41_{6.93}$ | $98.55_{1.44}$ |
|  | Action Executability | $93.14_{2.50}$ | $79.12_{4.26}$ | $94.12_{2.33}$ | $94.74_{3.62}$ |
|  | Effects of Actions | $78.49_{4.26}$ | $70.89_{5.11}$ | $77.17_{4.38}$ | $96.55_{2.40}$ |
|  | Numerical RAC | $62.82_{5.47}$ | $42.11_{8.01}$ | $57.38_{6.33}$ | $49.37_{5.62}$ |
|  | Composite Questions | $74.56_{4.08}$ | $50.43_{4.62}$ | $67.39_{3.99}$ | $94.42_{1.57}$ |
|  | Average | $81.26_{1.71}$ | $66.43_{2.30}$ | $77.76_{1.85}$ | $87.76_{1.43}$ |
| 10 | Fluent Tracking | $91.43_{3.35}$ | $68.42_{6.16}$ | $84.93_{4.19}$ | $89.86_{3.63}$ |
|  | State Tracking | $75.86_{5.62}$ | $76.19_{9.29}$ | $86.21_{6.40}$ | $96.43_{2.48}$ |
|  | Action Executability | $70.87_{4.48}$ | $65.67_{5.80}$ | $74.49_{4.40}$ | $81.82_{8.22}$ |
|  | Effects of Actions | $61.22_{4.92}$ | $63.86_{5.27}$ | $60.82_{4.96}$ | $96.67_{2.32}$ |
|  | Numerical RAC | $55.21_{5.08}$ | $65.52_{8.83}$ | $54.39_{6.60}$ | $51.14_{5.33}$ |
|  | Composite Questions | $82.96_{3.24}$ | $71.00_{4.54}$ | $68.92_{3.80}$ | $88.79_{2.07}$ |
|  | Average | $72.50_{1.89}$ | $67.79_{2.47}$ | $70.12_{2.04}$ | $84.06_{1.59}$ |
| 19 | Fluent Tracking | $91.53_{3.63}$ | $67.35_{6.70}$ | $77.27_{5.16}$ | $85.71_{4.68}$ |
|  | State Tracking | $85.94_{4.35}$ | $83.33_{8.78}$ | $81.82_{5.81}$ | $96.36_{2.52}$ |
|  | Action Executability | $66.67_{4.60}$ | $53.66_{7.79}$ | $66.99_{4.63}$ | $77.27_{6.32}$ |
|  | Effects of Actions | $70.00_{4.83}$ | $54.69_{6.22}$ | $69.32_{4.92}$ | $97.10_{2.02}$ |
|  | Numerical RAC | $50.00_{5.21}$ | $42.42_{8.60}$ | $47.54_{6.39}$ | $66.29_{5.01}$ |
|  | Composite Questions | $78.42_{3.49}$ | $53.09_{5.54}$ | $68.97_{3.84}$ | $86.18_{2.20}$ |
|  | Average | $72.31_{1.91}$ | $56.64_{2.93}$ | $68.24_{2.07}$ | $84.62_{1.53}$ |

Table 5: Performance comparison of GPT-4o, Llama-3.1-8B-Instruct, Llama-3.1-70B-Instruct, and fine-tuned Llama-3.1-8B on the binary subset (True/False) of the benchmark, evaluated without the ramifications constraints using the zero-shot-CoT. The results are categorized up by the action-sequence lengths and question categories.

## C.2 FEW-SHOT-3 RESULTS

Table 6 and 7 presents the results using the Few-shot-3 setting. These tables support the results showed in Section 5.

## C.3 RESULTS BY DOMAINS

Table 8 shows the results of the binary questions across every domain in our benchmark.

## C.4 RESULTS BY FLUENTS

Tables 9, 10 show the results of the binary questions across fluent types in our benchmark.

## D DATA VERIFICATION

To ensure the soundness of our benchmark, we employed three independent annotators who had no prior involvement with the project. Their task was to evaluate the naturalness of the sentences by assigning a score from 1 to 5, where 1 indicates the least natural and 5 most natural. To make sure that rephrasing the templated questions using Llama-3.1-70B-Instruct helps, we sample from each domain in the dataset was represented by 5 randomly sampled instances, resulting in a total of 65 samples across all domains for both templated questions and rephrased questions, resulting in a total of 130 samples. The annotators were provided with both the sampled instances and the following instructions:

> **Instruction to the Annotators**
>
> Rate the Prompts from 1 to 5, based on how natural they appear in English.

Table 11 summarizes the naturalness scores assigned by annotators across all domains in ACTION-REASONINGBENCH. The templated dataset received an average naturalness score of 4.2 out of

| Action Seq. | Ques Categories | GPT-4o | Llama-8B-Inst | Llama-70B-Inst |
|---|---|---|---|---|
| 1 | Fluent Tracking | $85.00_{3.57}$ | $68.42_{4.77}$ | $82.65_{3.82}$ |
| | State Tracking | $91.57_{3.05}$ | $71.95_{4.96}$ | $89.29_{3.37}$ |
| | Action Executability | $95.10_{2.14}$ | $84.31_{3.60}$ | $93.14_{2.50}$ |
| | Effects of Actions | $74.19_{4.54}$ | $73.91_{4.58}$ | $69.89_{4.76}$ |
| | Numerical RAC | $58.54_{5.44}$ | $56.76_{5.76}$ | $56.10_{5.48}$ |
| | Composite Questions | $82.66_{2.40}$ | $58.37_{3.23}$ | $74.30_{2.77}$ |
| | Average | $81.92_{1.45}$ | $67.26_{1.80}$ | $77.26_{1.58}$ |
| 10 | Fluent Tracking | $84.54_{3.67}$ | $69.66_{4.87}$ | $82.47_{3.86}$ |
| | State Tracking | $91.55_{3.30}$ | $64.18_{5.86}$ | $86.11_{4.08}$ |
| | Action Executability | $73.79_{4.33}$ | $63.00_{4.83}$ | $74.76_{4.28}$ |
| | Effects of Actions | $59.18_{4.96}$ | $53.68_{5.12}$ | $65.31_{4.81}$ |
| | Numerical RAC | $41.67_{5.03}$ | $42.25_{5.86}$ | $46.88_{5.09}$ |
| | Composite Questions | $82.96_{3.24}$ | $71.00_{4.54}$ | $68.92_{3.80}$ |
| | Average | $68.96_{1.71}$ | $59.51_{1.92}$ | $67.35_{1.74}$ |
| 19 | Fluent Tracking | $73.26_{4.77}$ | $63.38_{5.72}$ | $71.08_{4.98}$ |
| | State Tracking | $91.01_{3.03}$ | $63.64_{5.13}$ | $84.27_{3.86}$ |
| | Action Executability | $62.86_{4.72}$ | $54.55_{5.00}$ | $62.86_{4.72}$ |
| | Effects of Actions | $69.57_{4.80}$ | $56.63_{5.44}$ | $66.67_{4.89}$ |
| | Numerical RAC | $56.99_{5.13}$ | $49.15_{6.51}$ | $48.39_{5.18}$ |
| | Composite Questions | $65.57_{2.88}$ | $57.31_{3.11}$ | $62.04_{2.93}$ |
| | Average | $68.56_{1.71}$ | $57.58_{1.93}$ | $64.72_{1.76}$ |

Table 6: Performance comparison of GPT-4o, Llama-3.1-8B-Instruct, and Llama-3.1-70B-Instruct, on the binary subset (True/False) of the benchmark, evaluated without the ramifications constraints using the Few-shot-3 setting. The results are categorized up by the action-sequence lengths and question categories.

5, while the paraphrased version scored 4.5, indicating the high effectiveness of Llama-3.1-70B-Instruct in enhancing the fluency of the data. We would like to point out that the annotators participated on a voluntary basis and were informed beforehand that no financial compensation would be provided for their contribution.

## E  FINE-TUNING DETAILS

In this section, we describe the fine-tuning performed on the training split of ACTIONREASONING-BENCH described in section 3.5. We fine-tuned Llama-3.1-8B separately for binary (true/false) and free answer questions, using 6 epochs for the former and 18 epochs for the latter. The AdamW optimizer was used, with a batch size of 4 and gradient accumulation steps set to 8 for both of the training setups. Due to the available compute resources, we were limited to a maximum context length of 4096 tokens. This leaves us roughly with 27k samples for binary answers and 14.4k samples for free answer. Tables 12, 13 and 14 show the statistics of the training set that we used to train the models.

| Action Seq. | Ques Categories | GPT-4o | Llama-8B-Inst | Llama-70B-Inst |
|---|---|---|---|---|
| 1 | Fluent Tracking | $76.92_{5.84}$ | $50.00_{6.93}$ | $75.00_{6.00}$ |
| | State Tracking | $73.33_{6.59}$ | $44.44_{7.41}$ | $73.68_{7.14}$ |
| | Action Executability | $56.25_{7.16}$ | $14.58_{5.09}$ | $38.30_{7.09}$ |
| | Effects of Actions | $80.00_{6.32}$ | $35.00_{7.54}$ | $63.64_{8.37}$ |
| | Numerical RAC | $08.89_{4.24}$ | $04.44_{3.07}$ | $13.33_{5.07}$ |
| | Composite Questions | $49.26_{3.51}$ | $41.38_{3.46}$ | $55.61_{3.63}$ |
| | Average | $54.50_{2.39}$ | $35.33_{2.30}$ | $53.73_{2.49}$ |
| 10 | Fluent Tracking | $74.00_{6.20}$ | $24.00_{6.04}$ | $64.00_{6.79}$ |
| | State Tracking | $81.40_{5.93}$ | $34.88_{7.27}$ | $65.71_{8.02}$ |
| | Action Executability | $50.00_{7.54}$ | $09.09_{4.33}$ | $34.09_{7.15}$ |
| | Effects of Actions | $65.22_{7.02}$ | $41.30_{7.26}$ | $62.86_{8.17}$ |
| | Numerical RAC | $14.29_{5.00}$ | $10.20_{4.32}$ | $12.24_{4.68}$ |
| | Composite Questions | $43.84_{3.48}$ | $37.93_{3.41}$ | $48.13_{3.65}$ |
| | Average | $50.57_{2.40}$ | $30.34_{2.20}$ | $47.00_{2.50}$ |
| 19 | Fluent Tracking | $74.42_{6.65}$ | $44.19_{7.57}$ | $65.12_{7.27}$ |
| | State Tracking | $61.22_{6.96}$ | $28.57_{6.45}$ | $54.55_{7.51}$ |
| | Action Executability | $47.92_{7.21}$ | $10.42_{4.41}$ | $25.00_{6.25}$ |
| | Effects of Actions | $73.91_{6.47}$ | $27.66_{6.52}$ | $60.98_{7.62}$ |
| | Numerical RAC | $06.12_{3.42}$ | $06.12_{3.42}$ | $00.00_{0.00}$ |
| | Composite Questions | $38.19_{3.44}$ | $26.13_{3.11}$ | $46.2_{3.68}$ |
| | Average | $45.62_{2.39}$ | $24.37_{2.06}$ | $42.54_{2.44}$ |

Table 7: Performance comparison of GPT-4o, Llama-3.1-8B-Instruct, and Llama-3.1-70B-Instruct, on the free answer subset of the benchmark, evaluated without the ramifications constraints using the Few-shot-3 setting. The results are categorized up by the action-sequence lengths and question categories.

| Act. Seq. | Ques Categories | Blocksworld | Depots | Driverlog | Grippers | Mystery | Satellite | Spanner | Visitall |
|---|---|---|---|---|---|---|---|---|---|
| 1 | Fluent Tracking | $100.0_{0.0}$ | $92.31_{7.69}$ | $69.23_{13.32}$ | $95.24_{4.76}$ | $100.0_{0.0}$ | $88.24_{8.05}$ | $88.89_{7.62}$ | $100.0_{0.0}$ |
| | State Tracking | $83.33_{9.04}$ | $93.33_{6.67}$ | $66.67_{16.67}$ | $93.75_{6.25}$ | $85.71_{9.71}$ | $61.54_{14.04}$ | $62.5_{12.5}$ | $100.0_{0.0}$ |
| | Action Executability | $71.43_{10.1}$ | $76.47_{10.6}$ | $76.47_{10.6}$ | $78.95_{9.61}$ | $66.67_{11.43}$ | $76.19_{9.52}$ | $80.0_{9.18}$ | $47.06_{12.48}$ |
| | Effects of Act. | $82.61_{8.08}$ | $66.67_{12.6}$ | $85.71_{9.71}$ | $77.27_{9.14}$ | $100.0_{0.0}$ | $60.0_{13.09}$ | $80.0_{10.69}$ | $87.5_{8.54}$ |
| | Numerical RAC | $64.71_{11.95}$ | $53.85_{14.39}$ | $46.67_{13.33}$ | $50.0_{13.87}$ | $35.71_{13.29}$ | $44.44_{17.57}$ | $31.58_{10.96}$ | $31.82_{10.16}$ |
| | Composite | $78.57_{6.41}$ | $67.5_{7.5}$ | $68.42_{7.64}$ | $70.73_{7.19}$ | $66.67_{7.65}$ | $53.85_{8.09}$ | $60.98_{7.71}$ | $78.38_{6.86}$ |
| | Average | $78.95_{3.55}$ | $73.45_{4.17}$ | $68.87_{4.52}$ | $77.44_{3.64}$ | $73.21_{4.2}$ | $64.04_{4.51}$ | $65.89_{4.19}$ | $70.69_{4.24}$ |
| 10 | Fluent Tracking | $75.0_{25.0}$ | $91.67_{8.33}$ | $100.0_{0.0}$ | $100.0_{0.0}$ | $80.0_{13.33}$ | $73.68_{10.38}$ | $76.19_{9.52}$ | $94.74_{5.26}$ |
| | State Tracking | $86.67_{9.09}$ | $63.64_{15.21}$ | $83.33_{11.24}$ | $73.33_{11.82}$ | $63.64_{15.21}$ | $61.54_{14.04}$ | $81.25_{10.08}$ | $87.5_{12.5}$ |
| | Action Executability | $52.38_{11.17}$ | $64.29_{13.29}$ | $73.68_{10.38}$ | $57.14_{11.07}$ | $46.67_{13.33}$ | $64.71_{11.95}$ | $65.0_{10.94}$ | $55.0_{11.41}$ |
| | Effects of Act. | $68.0_{9.52}$ | $66.67_{10.54}$ | $78.57_{11.38}$ | $69.57_{9.81}$ | $78.57_{11.38}$ | $42.86_{11.07}$ | $61.54_{14.04}$ | $66.67_{10.54}$ |
| | Numerical RAC | $57.14_{11.07}$ | $44.0_{10.13}$ | $43.75_{12.81}$ | $37.5_{18.3}$ | $33.33_{12.6}$ | $42.86_{11.07}$ | $29.41_{11.39}$ | $27.27_{9.72}$ |
| | Composite | $71.74_{6.71}$ | $78.05_{6.54}$ | $74.42_{6.73}$ | $77.55_{6.02}$ | $55.56_{8.4}$ | $54.35_{7.43}$ | $64.29_{7.48}$ | $71.43_{7.75}$ |
| | Average | $67.42_{4.09}$ | $67.74_{4.21}$ | $74.36_{4.05}$ | $73.91_{3.75}$ | $57.43_{4.94}$ | $55.04_{4.4}$ | $63.57_{4.25}$ | $64.8_{4.29}$ |
| 19 | Fluent Tracking | $93.75_{6.25}$ | $66.67_{12.6}$ | $62.5_{18.3}$ | $100.0_{0.0}$ | $71.43_{18.44}$ | $93.33_{6.67}$ | $85.71_{9.71}$ | $61.54_{14.04}$ |
| | State Tracking | $91.67_{5.76}$ | $81.25_{10.08}$ | $86.67_{9.09}$ | $70.0_{15.28}$ | $87.5_{8.54}$ | $58.33_{14.86}$ | $78.57_{11.38}$ | $83.33_{16.67}$ |
| | Action Executability | $52.17_{10.65}$ | $64.29_{13.29}$ | $57.89_{11.64}$ | $70.0_{10.51}$ | $45.45_{10.87}$ | $56.25_{12.81}$ | $58.82_{12.3}$ | $68.18_{10.16}$ |
| | Effects of Act. | $72.73_{9.72}$ | $80.95_{8.78}$ | $85.71_{9.71}$ | $70.0_{10.51}$ | $76.92_{12.16}$ | $41.67_{14.86}$ | $66.67_{11.43}$ | $76.47_{10.6}$ |
| | Numerical RAC | $50.0_{11.47}$ | $57.89_{11.64}$ | $38.46_{14.04}$ | $30.0_{10.51}$ | $12.5_{8.54}$ | $37.5_{12.5}$ | $26.67_{11.82}$ | $31.82_{10.16}$ |
| | Composite | $72.92_{6.48}$ | $60.47_{7.54}$ | $72.09_{6.92}$ | $79.55_{6.15}$ | $53.85_{8.09}$ | $55.1_{7.18}$ | $67.44_{7.23}$ | $86.21_{6.52}$ |
| | Average | $71.9_{3.65}$ | $67.19_{4.17}$ | $68.75_{4.4}$ | $70.31_{4.05}$ | $54.87_{4.7}$ | $56.67_{4.54}$ | $64.46_{4.37}$ | $66.97_{4.53}$ |

Table 8: Performance across domains on GPT-4o on both the binary and free-answer subsets of the benchmark, evaluated without the ramifications constraints using the zero-shot setting. The results are categorized by the action-sequence lengths and question categories.

| Action Seq. | Fluent Types | GPT-4o | Llama-8B-Inst | Llama-70B-Inst | Finetuned Llama-8B |
|---|---|---|---|---|---|
| 1 | Base Fluents | $81.25_{5.63}$ | $65.22_{7.02}$ | $74.55_{5.87}$ | $96.3_{2.57}$ |
|  | Derived Fluents | $82.81_{4.72}$ | $76.36_{5.73}$ | $80.28_{4.72}$ | $86.75_{3.72}$ |
|  | Self-Derived Fluents | $85.87_{3.63}$ | $68.24_{5.05}$ | $81.44_{3.95}$ | $98.04_{1.37}$ |
|  | Static Properties | $85.19_{6.84}$ | $72.73_{7.75}$ | $80.0_{6.32}$ | $89.47_{4.06}$ |
| 10 | Base Fluents | $88.89_{4.68}$ | $78.79_{7.12}$ | $75.93_{5.82}$ | $90.57_{4.02}$ |
|  | Derived Fluents | $75.31_{4.79}$ | $55.38_{6.17}$ | $72.73_{5.08}$ | $79.38_{4.11}$ |
|  | Self-Derived Fluents | $77.66_{4.3}$ | $74.29_{5.22}$ | $69.39_{4.66}$ | $95.15_{2.12}$ |
|  | Static Properties | $79.41_{6.93}$ | $63.16_{7.83}$ | $69.77_{7.0}$ | $94.52_{2.66}$ |
| 19 | Base Fluents | $90.24_{4.63}$ | $63.64_{8.37}$ | $76.19_{6.57}$ | $83.61_{4.74}$ |
|  | Derived Fluents | $74.42_{4.7}$ | $44.07_{6.46}$ | $67.47_{5.14}$ | $76.47_{4.2}$ |
|  | Self-Derived Fluents | $80.0_{4.34}$ | $70.91_{6.12}$ | $71.74_{4.69}$ | $93.46_{2.39}$ |
|  | Static Properties | $82.76_{7.01}$ | $60.71_{9.23}$ | $67.5_{7.41}$ | $96.88_{2.17}$ |

Table 9: Performance comparison of GPT-4o, Llama-3.1-8B-Instruct, Llama-3.1-70B-Instruct, and fine-tuned Llama-3.1-8B on the binary subset (True/False) of the benchmark evaluated without the ramifications constraints using the zero-shot-CoT. The results are categorized up by the fluent types

| Action Seq. | Fluent Types | GPT-4o | Llama-8B-Inst | Llama-70B-Inst | Finetuned Llama-8B |
|---|---|---|---|---|---|
| 1 | Positive Fluents | $83.16_{3.84}$ | $80.0_{5.39}$ | $79.31_{4.34}$ | $72.84_{4.94}$ |
|  | Negative Fluents | $77.78_{4.62}$ | $60.34_{6.42}$ | $70.77_{5.64}$ | $77.53_{4.42}$ |
|  | Pos. and Neg. Fluents | $81.56_{2.08}$ | $65.15_{2.72}$ | $78.65_{2.17}$ | $93.77_{1.29}$ |
| 10 | Positive Fluents | $82.47_{3.86}$ | $80.77_{5.47}$ | $81.16_{4.71}$ | $80.0_{4.34}$ |
|  | Negative Fluents | $66.27_{5.19}$ | $53.66_{7.79}$ | $69.23_{5.72}$ | $72.29_{4.91}$ |
|  | Pos. and Neg. Fluents | $71.32_{2.32}$ | $67.42_{2.88}$ | $68.21_{2.43}$ | $87.74_{1.73}$ |
| 19 | Positive Fluents | $79.31_{4.34}$ | $67.44_{7.15}$ | $72.86_{5.32}$ | $84.93_{4.19}$ |
|  | Negative Fluents | $71.08_{4.98}$ | $58.14_{7.52}$ | $67.53_{5.34}$ | $75.64_{4.86}$ |
|  | Pos. and Neg. Fluents | $70.98_{2.33}$ | $54.0_{3.52}$ | $67.5_{2.47}$ | $86.27_{1.7}$ |

Table 10: Performance comparison of GPT-4o, Llama-3.1-8B-Instruct, Llama-3.1-70B-Instruct, and fine-tuned Llama-3.1-8B on the binary subset (True/False) of the benchmark evaluated without the ramifications constraints using the zero-shot-CoT. The results are categorized up by the fluent types

| Domain | Annotator 1 | | Annotator 2 | | Annotator 3 | | Average | |
|---|---|---|---|---|---|---|---|---|
|  | Templated | Rephrased | Templated | Rephrased | Templated | Rephrased | Templated | Rephrased |
| Blocksworld | 3.8 | 4.8 | 5.0 | 4.6 | 3.8 | 4.2 | 4.2 | 4.5 |
| Depots | 4.6 | 4.8 | 3.6 | 4.8 | 4.0 | 4.0 | 4.1 | 4.5 |
| Driverlog | 4.8 | 4.8 | 3.6 | 4.2 | 4.4 | 4.6 | 4.3 | 4.5 |
| Grippers | 4.6 | 5.0 | 4.0 | 4.8 | 4.6 | 4.8 | 4.4 | 4.9 |
| Mystery | 4.0 | 4.8 | 4.0 | 4.4 | 3.6 | 4.0 | 3.9 | 4.4 |
| Satellite | 4.4 | 4.8 | 4.6 | 4.4 | 4.4 | 4.2 | 4.5 | 4.5 |
| Spanner | 5.0 | 5.0 | 5.0 | 4.6 | 3.8 | 3.8 | 4.6 | 4.5 |
| Visitall | 3.6 | 4.2 | 3.8 | 4.6 | 4.0 | 4.2 | 3.8 | 4.3 |
| Average | 4.3 | 4.8 | 4.2 | 4.6 | 4.1 | 4.2 | 4.2 | 4.5 |

Table 11: Naturalness scores assigned by three annotators on a scale of 1 to 5, where 1 indicates completely incoherent text and 5 indicates natural-sounding questions. The table presents scores for both the templated questions and paraphrased questions.

| Answer category | No of Samples |
|---|---|
| False | 13,793 |
| True | 13,319 |
| Free-Response | 14,476 |
| Total | 41,588 |

Table 12: Data distribution used for fine-tuning, categorized by response type. The binary question responses are split into "True" and "False". "Free-Response" indicates the count of open-ended questions in the training set.

| Question category | No of Samples (Binary) | No of Samples(Free-Response) |
|---|---|---|
| Fluent Tracking | 11,674 | 4,946 |
| State Tracking | 2,264 | 1,133 |
| Action Executability | 1,534 | 1,094 |
| Effects of Actions | 1,196 | 1,040 |
| Numerical RAC | 5,757 | 3,293 |
| Composite Questions | 4,687 | 2,970 |
| Total | 27,112 | 14,476 |

Table 13: Data distribution used for fine-tuning, categorized by Question Categories.

| Domain | No of Samples (Binary) | No of Samples (Free-Response) |
|---|---|---|
| Blocksworld | 3,540 | 1,978 |
| Depots | 2,799 | 1,394 |
| Driverlog | 3,404 | 1,785 |
| Grippers | 3,494 | 1,910 |
| Mystery | 2,825 | 1,636 |
| Satellite | 4,037 | 2,073 |
| Spanner | 3,866 | 1,992 |
| Visitall | 3,147 | 1,708 |
| Total | 27,112 | 14,476 |

Table 14: Data distribution used for fine-tuning, categorized by Domains.

## F FREE ANSWERS EVALUATION DETAILS

We evaluate the free answers using Llama-3.1-70B-Instruct. The following is the few-shot-7 prompt that we used for evaluating the responses:

---

**Prompt for Free Answer Evaluation with Llama-3.1-70b-Instruct**

Evaluate whether the LLM response and the ground truth response are semantically the same. Examine the responses, provide reasoning for your evaluation, and then Write "True" if the responses are the same or "False" if they are different. LLM Response or Ground Truth could be "None".

Example 1:

[LLM Response]
Location f1_4f is connected to f1_3f, f0_4f, and f2_4f.

[Ground Truth]
location f1_4f and location f0_4f are connected, location f1_4f and location f2_4f are connected, location f0_4f and location f1_4f are connected, location f1_3f and location f1_4f are connected, there is a connection between location f2_4f and location f1_4f, there is a connection between location f1_4f and location f1_3f

[Reasoning for the evaluation]
all of the connections are the same

[Answer]
True

:

Example 7:

[LLM Response]
ball1 is at room2, ball2 is at room2, ball3 is at room4, ball4 is at room1, ball5 is at room1, ball6 is at room4, ball7 is being carried by robot1's lgripper1, ball8 is at room4, robot1 is at room2, robot1's lgripper1 is not free, robot1's rgripper1 is not free, robot1 is carrying ball7 with lgripper1, robot1 is carrying ball6 with rgripper1

[Ground Truth]
ball1 is present at room2, ball2 is at room2, ball3 is present at room4, ball4 is at room1, ball5 is located at room1, ball7 is being carried by robot1's lgripper1, ball8 is present at room4, rgripper1 of robot1 is carrying ball6 and robot1 is present in room2

[Reasoning for the evaluation]
According to the ground truth, robot1 is carrying ball6, but LLM gives a contradictory response: that ball6 is at room4

[Answer] False
_______________________________________________________________
Given the examples and instructions above, evaluate the following responses:

[LLM Response]
{llm_response}

[Ground Truth]
{ground_truth}

---

Table 15 shows the results of the free-answers using automatic metrics based on semantic similarity, F1-BERTScore (Zhang et al., 2019).

| Action Seq. | Ques Categories | GPT-4o | Llama-8B-Inst | Llama-70B-Inst | Finetuned Llama-8B |
|---|---|---|---|---|---|
| 1 | Fluent Tracking | $71.76_{2.11}$ | $48.52_{2.84}$ | $55.21_{2.85}$ | $90.22_{0.95}$ |
| | State Tracking | $68.24_{1.98}$ | $54.85_{3.18}$ | $61.07_{2.59}$ | $82.06_{1.82}$ |
| | Action Executability | $56.93_{3.36}$ | $49.49_{2.80}$ | $57.25_{3.77}$ | $75.24_{2.70}$ |
| | Effects of Actions | $66.43_{1.94}$ | $49.82_{3.53}$ | $59.48_{2.67}$ | $82.12_{2.07}$ |
| | Numerical RAC | $87.07_{1.61}$ | $39.40_{5.05}$ | $61.94_{5.30}$ | $89.52_{1.36}$ |
| | Composite Questions | $55.03_{1.65}$ | $40.59_{1.52}$ | $40.62_{1.77}$ | $82.58_{1.17}$ |
| | Average | $67.55_{2.10}$ | $47.11_{3.15}$ | $55.92_{3.15}$ | $83.62_{1.67}$ |
| 10 | Fluent Tracking | $66.46_{2.29}$ | $51.24_{2.98}$ | $56.89_{3.20}$ | $85.40_{1.74}$ |
| | State Tracking | $70.09_{1.26}$ | $60.43_{2.52}$ | $64.10_{2.23}$ | $83.78_{1.49}$ |
| | Action Executability | $61.31_{3.51}$ | $54.14_{2.79}$ | $68.28_{2.78}$ | $79.04_{2.40}$ |
| | Effects of Actions | $66.19_{1.34}$ | $52.99_{3.09}$ | $62.05_{2.16}$ | $82.03_{1.78}$ |
| | Numerical RAC | $82.38_{1.82}$ | $29.14_{4.13}$ | $50.39_{5.46}$ | $85.20_{2.62}$ |
| | Composite Questions | $54.57_{1.68}$ | $45.67_{1.59}$ | $54.41_{1.39}$ | $79.83_{1.56}$ |
| | Average | $66.83_{1.98}$ | $48.93_{2.85}$ | $59.35_{2.87}$ | $82.47_{1.93}$ |
| 19 | Fluent Tracking | $60.78_{3.19}$ | $47.69_{2.71}$ | $56.35_{3.12}$ | $87.79_{1.39}$ |
| | State Tracking | $70.33_{1.00}$ | $58.06_{2.77}$ | $62.06_{2.47}$ | $83.51_{1.41}$ |
| | Action Executability | $61.72_{3.41}$ | $49.40_{2.93}$ | $56.90_{3.48}$ | $78.84_{2.40}$ |
| | Effects of Actions | $65.71_{1.99}$ | $52.53_{3.51}$ | $62.78_{3.22}$ | $84.90_{1.20}$ |
| | Numerical RAC | $81.30_{2.29}$ | $32.55_{4.31}$ | $53.79_{5.36}$ | $80.75_{3.62}$ |
| | Composite Questions | $55.48_{1.55}$ | $50.87_{1.40}$ | $52.75_{1.63}$ | $77.10_{1.76}$ |
| | Average | $65.88_{2.23}$ | $48.51_{1.95}$ | $57.43_{3.21}$ | $82.14_{1.96}$ |

Table 15: F1-BERTScore of GPT-4o, Llama-3.1-8B-Instruct, Llama-3.1-70B-Instruct, and fine-tuned Llama-3.1-8B on the free-answer subset of the benchmark, evaluated without the ramifications constraints using the zero-shot-CoT. The results are categorized up by the action-sequence lengths and question categories.

Below we share some examples where F1-BertScore fails to give an accurate score.

---

**Low F1-BERTScore but the response is correct**

[GROUND TRUTH]
robot1's rgripper1 is available

[RESPONSE]
rgripper1 is free.

[BERT SCORE]
0.71

[Llama-3.1-70B-Intruct Evaluation]
The LLM response and the ground truth response are semantically the same. The term "free" in the LLM response is equivalent to "available" in the ground truth response, both indicating that the rgripper1 is not being used. True.

[ANALYSIS]
The response is a simple paraphrase of the ground truth which is easily explained by the LLM evaluation.

---

**High F1-BERTScore but the response is incorrect**

[GROUND TRUTH]
loc_x0_y1 is not connected to loc_x2_y4
loc_x0_y3 is not connected to loc_x2_y4
loc_x1_y0 and loc_x2_y4 are not connected
loc_x1_y1 is not connected to loc_x2_y4

⋮

there is no connection between loc_x3_y0 and loc_x2_y4
there is no connection between loc_x3_y1 and loc_x2_y4
there is no connection between loc_x3_y2 and loc_x2_y4
there is no connection between loc_x3_y3 and loc_x2_y4

[RESPONSE]
- loc_x2_y4 is not visited.
- loc_x2_y4 is connected to loc_x1_y4.
- loc_x2_y4 is connected to loc_x3_y4.
- loc_x2_y4 is not connected to loc_x2_y3.
- loc_x2_y4 is not connected to loc_x2_y5.

⋮

- loc_x2_y4 is not connected to loc_x1_y2.
- loc_x2_y4 is not connected to loc_x1_y3.
- loc_x2_y4 is not connected to loc_x1_y5.

[BERT SCORE]
0.89

[Llama-3.1-70B-Intruct Evaluation]
The LLM response and the ground truth response are not semantically the same. The LLM response states that loc_x2_y4 is not connected to loc_x2_y3, but the ground truth response states that loc_x2_y4 and loc_x2_y3 are connected. Additionally, the LLM response states that loc_x2_y4 is not connected to loc_x2_y5, but the ground truth response states that loc_x2_y4 and loc_x2_y5 are connected. False.

[ANALYSIS]
The response contains a lot of incorrect responses. Moreover, the response fails to mention the location of the robot. The Llama-3.1-70B-Instruct evaluation also fails to mention the lack of the robot but correctly identifies some incorrect connections.

---

# G  DOMAINS IN ACTIONREASONINGBENCH

In the following sub-sections, we provide details regarding all the domains used in our study. We first present the PDDL description that is given in the IPC and then present how state-space calculation is performed for that domain. State space represents the possible number of interactions that can be performed at a particular state. A higher state-space represents a more difficult problem for traditional AI solvers.

## G.1  BLOCKSWORLD

In this domain, we have a set of blocks that can be manipulated using four basic actions: picking up a block from the table, putting down a block onto the table, stacking one block onto another, and unstacking a block from atop another block. The goal is to move and stack these blocks using a robotic hand, following specific rules and conditions.

This domain is formally represented in the IPC using the PDDL as outlined below:

| Domain Name | # Fluents Defined | # Actions Defined | # Object Defined | IPC Year | State Complexity |
|---|---|---|---|---|---|
| **Blocksworld** | 5 | 4 | 1 | 2000 | $O(2^{N^2+2N})$ |
| **Depots** | 6 | 5 | 6 | 2002 | $O(2^{4N^2+2N})$ |
| **DriverLog** | 5 | 6 | 4 | 2002 | $O(2^{5N^2-N})$ |
| **Grippers** | 4 | 3 | 4 | 1998 | $O(2^{N^3+3N^2})$ |
| **Mystery** | 7 | 3 | 5 | 1998 | $O(2^{7N^2-3N})$ |
| **Satellite** | 8 | 5 | 4 | 2002 | $O(2^{5N^2+3N})$ |
| **Spanner** | 6 | 3 | 4 | 2011 | $O(2^{3N^2+2N})$ |
| **Visitall** | 3 | 1 | 1 | 2014 | $O(2^{N^2+N})$ |

Table 16: Summarizing key characteristics of various domains in ACTIONREASONINGBENCH including the number of fluents, actions, and objects defined within each domain. Domains are categorized by year of introduction in the IPC and state space complexity, which reflects the difficulty level for AI planners to solve each domain. A larger state space typically indicates greater complexity and presents more significant challenges for traditional AI planners. "$N$" represents the number of objects in each instance. For example, in the *Spanner* domain, $N$ refers to the number of locations, spanners and nuts.

---

PDDL description for `Blocksworld` domain

```
(define (domain BLOCKS)
(:requirements :strips :typing)
(:types block)
(:predicates (on ?x - block ?y - block)
(ontable ?x - block)
(clear ?x - block)
(handempty)
(holding ?x - block))

(:action pick-up
:parameters (?x - block)
:precondition (and (clear ?x) (ontable ?x) (handempty))
:effect
(and (not (ontable ?x))
(not (clear ?x))
(not (handempty))
(holding ?x)))
```

```
(:action put-down
:parameters (?x - block)
:precondition (holding ?x)
:effect
(and (not (holding ?x))
(clear ?x)
(handempty)
(ontable ?x)))

(:action stack
:parameters (?x - block ?y - block)
:precondition (and (holding ?x) (clear ?y))
:effect
(and (not (holding ?x))
(not (clear ?y))
(clear ?x)
(handempty)
(on ?x ?y)))

(:action unstack
:parameters (?x - block ?y - block)
:precondition (and (on ?x ?y) (clear ?x) (handempty))
:effect
(and (holding ?x)
(clear ?y)
(not (clear ?x))
(not (handempty))
(not (on ?x ?y))))
)
```

Predicates define the number of fluents in a state:

- on(b1,b2)
- ontable(b)
- clear(b)
- holding(b)
- handempty

The complexity of a state is, where $N$ is the number of objects

$$O(2^{N^2+2N}) \tag{1}$$

## G.2 DEPOTS

The Depots domain models a logistics environment where crates are transported between different locations using trucks and manipulated using hoists. The goal is to efficiently move crates from one location to another, utilizing the available resources (hoists and trucks) while adhering to the constraints defined by the predicates and actions.

This domain is formally represented in the IPC using the PDDL as outlined below:

PDDL description for `Depots` domain

```
(define (domain depots)
(:requirements :strips :typing)
(:types place locatable - object
depot distributor - place
```

```
truck hoist surface - locatable
pallet crate - surface)

(:predicates (at ?x - locatable ?y - place)
(on ?x - crate ?y - surface)
(in ?x - crate ?y - truck)
(lifting ?x - hoist ?y - crate)
(available ?x - hoist)
(clear ?x - surface))

(:action Drive
:parameters (?x - truck ?y - place ?z - place)
:precondition (and (at ?x ?y))
:effect (and (not (at ?x ?y)) (at ?x ?z)))

(:action Lift
:parameters (?x - hoist ?y - crate ?z - surface ?p - place)
:precondition (and (at ?x ?p) (available ?x) (at ?y ?p) (on ?y ?z) (clear ?y))
:effect (and (not (at ?y ?p)) (lifting ?x ?y) (not (clear ?y)) (not (available ?x)) (clear ?z) (not
(on ?y ?z))))

(:action Drop
:parameters (?x - hoist ?y - crate ?z - surface ?p - place)
:precondition (and (at ?x ?p) (at ?z ?p) (clear ?z) (lifting ?x ?y))
:effect (and (available ?x) (not (lifting ?x ?y)) (at ?y ?p) (not (clear ?z)) (clear ?y)(on ?y
?z)))

(:action Load
:parameters (?x - hoist ?y - crate ?z - truck ?p - place)
:precondition (and (at ?x ?p) (at ?z ?p) (lifting ?x ?y))
:effect (and (not (lifting ?x ?y)) (in ?y ?z) (available ?x)))

(:action Unload
:parameters (?x - hoist ?y - crate ?z - truck ?p - place)
:precondition (and (at ?x ?p) (at ?z ?p) (available ?x) (in ?y ?z))
:effect (and (not (in ?y ?z)) (not (available ?x)) (lifting ?x ?y)))
)
```

Predicates define the number of fluents in a state:

- (at ?x - locatable ?y - place)
- (on ?x - crate ?y - surface)
- (in ?x - crate ?y - truck)
- (lifting ?x - hoist ?y - crate)
- (available ?x - hoist)
- (clear ?x - surface))

$$O(2^{4N^2+2N}) \tag{2}$$

## G.3 DRIVERLOG

This domain is modeled to simulate logistics operations where drivers, trucks, and objects must be moved between different locations. The primary focus is on transporting objects via trucks, either driven by drivers or moved manually by walking. The key actions in this domain include loading and unloading trucks, drivers boarding and disembarking trucks, driving trucks between connected locations, and walking when no truck is involved.

This domain is formally represented in the IPC using the PDDL as outlined below:

---

### PDDL description for `Driverlog` domain

```
(define (domain driverlog)
(:requirements :typing) (:types location locatable - object
driver truck obj - locatable)
(:predicates
(at ?obj - locatable ?loc - location)
(in ?obj1 - obj ?obj - truck)
(driving ?d - driver ?v - truck)
(link ?x ?y - location) (path ?x ?y - location)
(empty ?v - truck))

(:action LOAD-TRUCK
:parameters
(?obj - obj
?truck - truck
?loc - location)
:precondition (and (at ?truck ?loc) (at ?obj ?loc))
:effect (and (not (at ?obj ?loc)) (in ?obj ?truck)))

(:action UNLOAD-TRUCK
:parameters
(?obj - obj
?truck - truck
?loc - location)
:precondition (and (at ?truck ?loc) (in ?obj ?truck))
:effect (and (not (in ?obj ?truck)) (at ?obj ?loc)))

(:action BOARD-TRUCK
:parameters
(?driver - driver
?truck - truck
?loc - location)
:precondition (and (at ?truck ?loc) (at ?driver ?loc) (empty ?truck))
:effect (and (not (at ?driver ?loc)) (driving ?driver ?truck) (not (empty ?truck))))

(:action DISEMBARK-TRUCK
:parameters
(?driver - driver
?truck - truck
?loc - location)
:precondition (and (at ?truck ?loc) (driving ?driver ?truck))
:effect (and (not (driving ?driver ?truck)) (at ?driver ?loc) (empty ?truck)))

(:action DRIVE-TRUCK
:parameters
(?truck - truck
?loc-from - location
?loc-to - location
?driver - driver)
:precondition
(and (at ?truck ?loc-from)
(driving ?driver ?truck) (link ?loc-from ?loc-to))
:effect (and (not (at ?truck ?loc-from)) (at ?truck ?loc-to)))
```

---

```
(:action WALK
:parameters
(?driver - driver
?loc-from - location
?loc-to - location)
:precondition (and (at ?driver ?loc-from) (path ?loc-from ?loc-to))
:effect (and (not (at ?driver ?loc-from)) (at ?driver ?loc-to)))
)
```

Predicates define the number of fluents in a state:

- (at ?obj - locatable ?loc - location)

- (in ?obj1 - obj ?obj - truck)

- (driving ?d - driver ?v - truck)

- (link ?x ?y - location)

- (path ?x ?y - location)

- (empty ?v - truck)

$$O(2^{5N^2 - N}) \tag{3}$$

## G.4 GRIPPER

This domain represents a transportation domain where a robot with two grippers can move between rooms, pick up objects, and drop them off. The robot can only hold one object in each gripper at a time. This domain could solve tasks where the robot needs to transport multiple objects from one room to another by strategically moving, picking up, and dropping items.

This domain is formally represented in the IPC using the PDDL as outlined below:

PDDL description for `Gripper` domain

```
(define (domain gripper-strips)
(:requirements :strips :typing)
(:types room object robot gripper)
(:predicates (at-robby ?r - robot ?x - room)
(at ?o - object ?x - room)
(free ?r - robot ?g - gripper)
(carry ?r - robot ?o - object ?g - gripper))

(:action move
:parameters (?r - robot ?from ?to - room)
:precondition (and (at-robby ?r ?from))
:effect (and (at-robby ?r ?to) (not (at-robby ?r ?from))))

(:action pick
:parameters (?r - robot ?obj - object ?room - room ?g - gripper)
:precondition (and (at ?obj ?room) (at-robby ?r ?room) (free ?r ?g))
:effect (and (carry ?r ?obj ?g)
(not (at ?obj ?room))
(not (free ?r ?g))))
```

```
(:action drop
:parameters (?r - robot ?obj - object ?room - room ?g - gripper)
:precondition (and (carry ?r ?obj ?g) (at-robby ?r ?room))
:effect (and (at ?obj ?room)
(free ?r ?g)
(not (carry ?r ?obj ?g)))))
```

Predicates define the number of fluents in a state:

- (carry ?r - robot ?o - object ?g - gripper)

- (at-robby ?r - robot ?x - room)

- (at ?o - object ?x - room)

- (free ?r - robot ?g - gripper)

$$O(2^{N^3 + 3N^2}) \tag{4}$$

## G.5 MYSTERY

The Mystery domain represents a transportation system where vehicles move between locations, constrained by fuel levels, and can load or unload cargo, constrained by available space. The key aspects of this domain are managing fuel for vehicle movement and managing space for loading and unloading cargo. Locations are connected, and the system also handles fuel transitions, space transitions, and the movement of objects across a grid of locations.

This domain is formally represented in the IPC using the PDDL as outlined below:

PDDL description for `Mystery` domain

```
(define (domain mystery-strips)
(:requirements :typing)
(:types space fuel location movable - object
vehicle cargo - movable)
(:predicates
(at ?v - movable ?l - location)
(conn ?l1 ?l2 - location)
(has-fuel ?l - location ?f - fuel)
(fuel-neighbor ?f1 ?f2 - fuel)
(in ?c - cargo ?v - vehicle)
(has-space ?v - vehicle ?s - space)
(space-neighbor ?s1 ?s2 - space))

(:action move
:parameters (?v - vehicle ?l1 ?l2 - location ?f1 ?f2 - fuel)
:precondition (and (at ?v ?l1)
(conn ?l1 ?l2)
(has-fuel ?l1 ?f1)
(fuel-neighbor ?f2 ?f1))
:effect (and (not (at ?v ?l1))
(at ?v ?l2)
(not (has-fuel ?l1 ?f1))
(has-fuel ?l1 ?f2)))
```

```
(:action load
:parameters (?c - cargo ?v - vehicle ?l - location ?s1 ?s2 - space)
:precondition (and (at ?c ?l)
(at ?v ?l)
(has-space ?v ?s1)
(space-neighbor ?s2 ?s1))
:effect (and (not (at ?c ?l))
(in ?c ?v)
(not (has-space ?v ?s1))
(has-space ?v ?s2)))

(:action unload
:parameters (?c - cargo ?v - vehicle ?l - location ?s1 ?s2 - space)
:precondition (and (in ?c ?v)
(at ?v ?l)
(has-space ?v ?s1)
(space-neighbor ?s1 ?s2))
:effect (and (not (in ?c ?v))
(at ?c ?l)
(not (has-space ?v ?s1))
(has-space ?v ?s2)))
)
```

Predicates define the number of fluents in a state:

- (at ?v - movable ?l - location)

- (has-fuel ?l - location ?f - fuel)

- (in ?c - cargo ?v - vehicle)

- (has-space ?v - vehicle ?s - space)

- (conn ?l1 ?l2 - location)

- (fuel-neighbor ?f1 ?f2 - fuel)

- (space-neighbor ?s1 ?s2 - space)

$$O(2^{7N^2-3N}) \tag{5}$$

### G.6 SATELLITE

The Satellite domain represents a simplified model for managing and controlling satellites and their onboard instruments. The goal in this domain is to coordinate the behavior of satellites, including turning them toward desired directions, powering instruments on and off, calibrating instruments, and capturing images.

This domain is formally represented in the IPC using the PDDL as outlined below:

PDDL description for `Satellite` domain

```
(define (domain satellite)
(:requirements :strips :typing)
(:types satellite direction instrument mode)
```

```
(:predicates
(on_board ?i - instrument ?s - satellite)
(supports ?i - instrument ?m - mode)
(pointing ?s - satellite ?d - direction)
(power_avail ?s - satellite)
(power_on ?i - instrument)
(calibrated ?i - instrument)
(have_image ?d - direction ?m - mode)
(calibration_target ?i - instrument ?d - direction))

(:action turn_to
:parameters (?s - satellite ?d_new - direction ?d_prev - direction)
:precondition (and (pointing ?s ?d_prev))
:effect (and (pointing ?s ?d_new) (not (pointing ?s ?d_prev))))

(:action switch_on
:parameters (?i - instrument ?s - satellite)
:precondition (and (on_board ?i ?s) (power_avail ?s))
:effect (and (power_on ?i) (not (calibrated ?i)) (not (power_avail ?s))))

(:action switch_off
:parameters (?i - instrument ?s - satellite)
:precondition (and (on_board ?i ?s) (power_on ?i))
:effect (and (not (power_on ?i)) (power_avail ?s)))

(:action calibrate
:parameters (?s - satellite ?i - instrument ?d - direction)
:precondition (and (on_board ?i ?s)
(calibration_target ?i ?d)
(pointing ?s ?d)
(power_on ?i))
:effect (calibrated ?i))

(:action take_image
:parameters (?s - satellite ?d - direction ?i - instrument ?m - mode)
:precondition (and (calibrated ?i)
(on_board ?i ?s)
(supports ?i ?m)
(power_on ?i)
(pointing ?s ?d))
:effect (have_image ?d ?m)))
```

Predicates define the number of fluents in a state:

- (on-board ?i - instrument ?s - satellite)
- (supports ?i - instrument ?m - mode)
- (pointing ?s - satellite ?d - direction)
- (have-image ?d - direction ?m - mode)
- (calibration-target ?i - instrument ?d - direction)
- (power-avail ?s - satellite)
- (power-on ?i - instrument)
- (calibrated ?i - instrument)

$$O(2^{5N^2+3N}) \tag{6}$$

## G.7 SPANNER

This domain models a simple world where a man moves between locations, picks up spanners, and uses them to tighten loose nuts. The actions available to the man involve walking between locations, picking up the spanner, and tightening nuts using the spanner if all conditions are met.

This domain is formally represented in the IPC using the PDDL as outlined below:

```
PDDL description for Spanner domain

(define (domain spanner)
(:requirements :typing :strips)
(:types
location locatable - object
man nut spanner - locatable
)

(:predicates
(at ?m - locatable ?l - location)
(carrying ?m - man ?s - spanner)
(useable ?s - spanner)
(link ?l1 - location ?l2 - location)
(tightened ?n - nut)
(loose ?n - nut))

(:action walk
:parameters (?start - location ?end - location ?m - man)
:precondition (and (at ?m ?start) (link ?start ?end))
:effect (and (not (at ?m ?start)) (at ?m ?end)))

(:action pickup_spanner
:parameters (?l - location ?s - spanner ?m - man)
:precondition (and (at ?m ?l) (at ?s ?l))
:effect (and (not (at ?s ?l)) (carrying ?m ?s)))

(:action tighten_nut
:parameters (?l - location ?s - spanner ?m - man ?n - nut)
:precondition (and (at ?m ?l)
(at ?n ?l)
(carrying ?m ?s)
(useable ?s)
(loose ?n))
:effect (and (not (loose ?n))(not (useable ?s)) (tightened ?n)))
)
```

Predicates define the number of fluents in a state:

- (at ?m - locatable ?l - location)

- (carrying ?m - man ?s - spanner)

- (link ?l1 - location ?l2 - location)

- (useable ?s - spanner)

- (tightened ?n - nut)

- (loose ?n - nut))

$$O(2^{3N^2+2N}) \tag{7}$$

## G.8   VISITALL

The VisitAll domain is focused on controlling a robot that needs to visit all places on a connected grid. The robot's movement is governed by the connectivity of places, and each move changes the robot's location and marks the visited place. The task is essentially to traverse the entire grid, visiting every place, while ensuring the robot follows connectivity constraints.

This domain is formally represented in the IPC using the PDDL as outlined below:

---
**PDDL description for `VisitAll` domain**

```
(define (domain grid-visit-all)
(:requirements :typing)
(:types place - object)
(:predicates (connected ?x ?y - place)
(at-robot ?x - place)
(visited ?x - place))

(:action move
:parameters (?curpos ?nextpos - place)
:precondition (and (at-robot ?curpos) (connected ?curpos ?nextpos))
:effect (and (at-robot ?nextpos) (not (at-robot ?curpos)) (visited ?nextpos)))
)
```
---

Predicates define the number of fluents in a state:

- (connected ?x ?y - place)
- (at-robot ?x - place)
- (visited ?x - place)

$$O(2^{N^2+N}) \tag{8}$$

## H   PLANNING DESCRIPTION AND TOOLS

Planning, at its core, involves determining a sequence of actions that transforms the world from an initial state to a goal state. A world state specifies which fluents are true or false at any given time. The planning domain, denoted as $D$, specifies the fluents, actions, and their effects within the system. Typically, planning domains are represented using formal languages such as PDDL or ASP. In these languages, a transition function $\Phi_D : states \times actions \rightarrow states$ defines how actions transform an initial state into a resulting state.

### H.1   PLANNING DOMAIN DEFINITION LANGUAGE (PDDL)

PDDL is a formal language developed for expressing planning problems and domain models. Since its inception, PDDL has been extended to address increasingly complex planning scenarios, particularly those involving deterministic problems (Haslum et al., 2019). PDDL facilitates the specification of both the planning domains and problem instances, including objects, initial, and goal states. In this study, we employ the "STRIPS" (Stanford Research Institute Problem Solver) subset of PDDL (Fikes & Nilsson, 1971). Additionally, the domains are "typed", meaning that objects in the planning problem are assigned specific types and subtypes, ensuring a structured representation of the problem space.

### H.2   ANSWER SET PROGRAMMING (ASP)

ASP is a declarative approach to problem-solving based on logic programming and non-monotonic reasoning. Unlike traditional planning languages like PDDL, ASP focuses on defining constraints and rules that describe potential solutions, rather than directly encoding state transitions. In ASP, a

problem is encoded as a logic program consisting of rules, facts, and constraints, and the solution is an "answer set" that satisfies all the constraints of the problem. In this study, we use ASP to generate the complete state-space by applying the sequence of actions starting from the initial state.

# I  CLASSIFICATION OF FLUENTS

The subsequent sections provide a detailed classification of fluents across all 13 domains included in ACTIONREASONINGBENCH, as described in section 3.2.

## I.1  BLOCKSWORLD

In the BLOCKSWORLD domain, the fluents are categorized as follows:

1. Static Properties - No static properties are present
2. Base Fleunts - `onTable(block)`
3. Derived Fluents - `clear(block)`, `handEmpty`
4. Self-Derived Fluents - `holding(block)`, `on(block,block)`

## I.2  DEPOTS

In the DEPOTS domain, the fluents are categorized as follows:

1. Static Properties - No static properties are present
2. Base Fleunts - No base fluents are present
3. Derived Fluents - `clear(surface)`, `available(hoist)`
4. Self-Derived Fluents - `at(locatable,place)`, `on(crate,surface)`, `in(crate,truck)`, `lifting(hoist,crate)`

## I.3  DRIVERLOG

In the DRIVERLOG domain, the fluents are categorized as follows:

1. Static Properties - `link(location,location)`, `path(location,location)`
2. Base Fleunts - No base fluents are present
3. Derived Fluents - `empty(truck)`
4. Self-Derived Fluents - `at(locatable,location)`, `in(object,truck)`, `driving(driver,truck)`

## I.4  GRIPPERS

In the GRIPPERS domain, the fluents are categorized as follows:

1. Static Properties - No static properties are present
2. Base Fleunts - `carry(robot,object,gripper)`
3. Derived Fluents - `free(robot,gripper)`
4. Self-Derived Fluents - `at_robby(robot,room)`, `at(object,room)`

## I.5  MYSTERY

In the MYSTERY domain, the fluents are categorized as follows:

1. Static Properties - `space_neighbor(space,space)`, `fuel_neighbor(fuel,fuel)`, `conn(location,location)`

2. Base Fleunts - No base fluents are present

3. Derived Fluents - No derived fluents are present

4. Self-Derived Fluents - `at(movable,location)`, `in(cargo,vehicle)`, `has_space(vehicle,space)`, `has_fuel(location,fuel)`

## I.6    SATELLITE

In the SATELLITE domain, the fluents are categorized as follows:

1. Static Properties - `on_board(instrument,satellite)`, `supports(instrument,mode)`, `calibration_target(instrument,direction)`

2. Base Fleunts - `power_on(instrument)`, `calibrated(instrument)`, `have_image(direction,mode)`

3. Derived Fluents - `power_avail(satellite)`

4. Self-Derived Fluents - `pointing(satellite,direction)`

## I.7    SPANNER

In the SPANNER domain, the fluents are categorized as follows:

1. Static Properties - `link(location,location)`

2. Base Fleunts - `carrying(man,spanner)`, `tightened(nut)`

3. Derived Fluents - `loose(nut)`, `useable(spanner)`

4. Self-Derived Fluents - `at(locatable,location)`

## I.8    VISITALL

In the VISITALL domain, the fluents are categorized as follows:

1. Static Properties - `connected(place,place)`

2. Base Fleunts - No base fluents are present

3. Derived Fluents - `visited(place)`

4. Self-Derived Fluents - `at_robot(place)`

## J    RESPOSNES ON RAMIFICATION DATASET

In the following sections, we illustrate some instances and responses that we see from GPT-4o and o1-preview.

## J.1 GPT-4O

---

**A Ramificaiton question for State Tracking**

[DOMAIN DESCRIPTION]
A block can only be picked up if it is clear, on the table, and the hand is empty, resulting in the block being held. A held block can be put down, placing it back on the table. Blocks can be stacked if the first block is held and the second block is clear, causing the first block to rest on top of the second. Unstacking occurs when the hand is empty, the first block is clear, and on top of the second, resulting in the first block being held again. A block is clear if nothing is on top of it and it's not held, and the hand is empty if it's not holding anything. Blocks are stable when clear and on the table, and they can be painted if stable and the hand is empty. A block is considered on display if it can be painted and has no other block on top of it.

[INITIAL CONDITIONS]
Block b1 is stacked on block b4, block b2 has no blocks on top of it, block b2 is stacked on block b6, block b3 has no blocks on top of it, block b3 is stacked on block b5, block b4 is positioned on the table, block b5 is stacked on block b7, block b6 is positioned on the table, block b7 is stacked on block b1 and the hand is empty.

[QUESTION]
Given the initial condition, the following actions are taken: block b3 is unstacked from block b5 to achieve the current state. In this state, if block b3 is placed down, what are all the valid properties of the state (including both affirmative and negated properties)? If there are no valid properties, state None.

---

**Response to the above question**

[REASON]
To determine the valid properties of the state after block b3 is placed down, we need to follow the sequence of actions and check the conditions for each property.

1. **Initial Conditions:**
- b1 is on b4.
- b2 is on b6.
- b3 is on b5.
- b4 is on the table.
- b5 is on b7.
- b6 is on the table.
- b7 is on b1.
- The hand is empty.

2. **Action: Unstack b3 from b5:**
- The hand must be empty.
- b3 must be clear and on top of b5.
- Result: b3 is held, b5 is now clear.

3. **Current State after unstacking b3:**
- b1 is on b4.
- b2 is on b6.
- b3 is held.

- b4 is on the table.
- b5 is on b7 and is clear.
- b6 is on the table.
- b7 is on b1.
- The hand is not empty (holding b3).

4. **Action: Place b3 down:**
- The hand must be holding b3.
- Result: b3 is placed on the table, the hand is empty.

5. **Final State after placing b3 down:**
- b1 is on b4.
- b2 is on b6.
- b3 is on the table and clear.
- b4 is on the table.
- b5 is on b7 and is clear.
- b6 is on the table.
- b7 is on b1.
- The hand is empty.

6. **Valid Properties:**
- b1 is on b4.
- b2 is on b6.
- b3 is on the table.
- b3 is clear.
- b4 is on the table.
- b5 is on b7.
- b5 is clear.
- b6 is on the table.
- b7 is on b1.
- The hand is empty.
- b1 is not clear (b7 is on it).
- b2 is not clear (nothing on it but it is on b6).
- b4 is clear (nothing on it and on the table).
- b6 is clear (nothing on it and on the table).
- b7 is not clear (b5 is on it).
- b3 is stable (clear and on the table).
- b4 is stable (clear and on the table).
- b6 is stable (clear and on the table).
- b3 can be painted (stable and hand is empty).
- b4 can be painted (stable and hand is empty).
- b6 can be painted (stable and hand is empty).
- b3 is on display (can be painted and no block on top).
- b4 is on display (can be painted and no block on top).
- b6 is on display (can be painted and no block on top).

[ANSWER]
- b1 is on b4.
- b2 is on b6.
- b3 is on the table.
- b3 is clear.
- b4 is on the table.
- b5 is on b7.
- b5 is clear.
- b6 is on the table.
- b7 is on b1.

- The hand is empty.
- b1 is not clear.
- b2 is not clear.
- b4 is clear.
- b6 is clear.
- b7 is not clear.
- b3 is stable.
- b4 is stable.
- b6 is stable.
- b3 can be painted.
- b4 can be painted.
- b6 can be painted.
- b3 is on display.
- b4 is on display.
- b6 is on display.

From the above response, we can see that block b4 is **not** stable, but the model says it is stable.

## J.2  O1-PREVIEW

**A Ramificaiton question for State Tracking**

[DOMAIN DESCRIPTION]
A block can only be picked up if it is clear, on the table, and the hand is empty, resulting in the block being held. A held block can be put down, placing it back on the table. Blocks can be stacked if the first block is held and the second block is clear, causing the first block to rest on top of the second. Unstacking occurs when the hand is empty, the first block is clear, and on top of the second, resulting in the first block being held again. A block is clear if nothing is on top of it and it's not held, and the hand is empty if it's not holding anything. Blocks are stable when clear and on the table, and they can be painted if stable and the hand is empty. A block is considered on display if it can be painted and has no other block on top of it.

[INITIAL CONDITIONS]
Block b1 is positioned above block b7, block b2 has no blocks on it, block b2 is stacked on block b6, block b3 is stacked on block b4, block b4 is situated on the table, block b5 has no blocks on it, block b5 is situated on the table, block b6 is stacked on block b3, block b7 is on the table, block b8 is positioned above block b1, block b9 has no blocks on it, block b9 is stacked on block b8, and the hand is empty.

[QUESTION]
Based on the initial condition, the following actions are taken: block b9 is removed from block b8 to achieve the current state. In this state, identify all valid properties that do not include negations and list them; if there are no such properties, indicate None.

---

**Response to the above question**

[REASON]
After removing block b9 from block b8, block b9 is placed on the table, and the hand remains empty. Block b9 now has no blocks on top of it and is not held, so it is **clear**. Since it is on the table, it is also **stable**. With the hand empty and block b9 being stable, it **can be painted**. As there are no blocks on top of it, block b9 is **on display**.

Similarly, block b5 is on the table with no blocks on top and is not held, making it **clear** and **stable**. It **can be painted** and is **on display**.

Block b7 is on the table and, with no blocks explicitly on top of it after the action, is **clear**, **stable**, **can be painted**, and is **on display**.

Blocks b1 and b8 have no blocks on top of them and are not held, so they are **clear**.

Block b2 has no blocks on top and is not held, so it is **clear**, though it is not on the table and thus not stable.

The hand is explicitly stated to be **empty** in the current state.

[ANSWER]
- The hand is empty
- Block b1 is clear
- Block b2 is clear
- Block b5 is clear
- Block b5 is on the table
- Block b5 is stable
- Block b5 can be painted
- Block b5 is on display
- Block b7 is clear
- Block b7 is on the table
- Block b7 is stable
- Block b7 can be painted
- Block b7 is on display
- Block b8 is clear
- Block b9 is clear
- Block b9 is on the table
- Block b9 is stable
- Block b9 can be painted
- Block b9 is on display

---

From the above response, we can see that block b1 is **not** clear, but the model says it is clear.

## K    IMPACT OF INPUT FORMATS ON LLM PERFORMANCE

In this section, we explore the effects of different input formats on LLMs. Specifically, we compare the performance of LLMs when inputs are presented in formal language (PDDL) or in templated formats generated during the data creation pipeline (described in Section 3.4), as opposed to natural language. For our analysis, we sampled 10% of the test set from ACTIONREASONINGBENCH, covering all question categories and action sequence lengths.

### K.1    FORMAL LANGUAGE (PDDL)

Table 17 presents the performance of LLMs when provided inputs in the formal language PDDL. Our results reveal a significant drop in performance for most models. GPT-4o shows a notable performance degradation of 16.03% compared to the natural language baseline. Similarly, Llama-3.1-70B-Instruct experiences a 6.7% decrease in accuracy. Interestingly, Llama-3.1-8B-Instruct exhibits

| Action Seq. | GPT-4o | Llama-8B-Inst | Llama-70B-Inst |
|:---:|:---:|:---:|:---:|
| 1 | $43.90_{7.75}$ | $26.83_{6.92}$ | $53.66_{7.79}$ |
| 10 | $48.78_{7.81}$ | $17.07_{5.88}$ | $31.71_{7.27}$ |
| 19 | $33.33_{7.55}$ | $17.95_{6.15}$ | $25.64_{6.99}$ |

Table 17: Performance comparison of GPT-4o, Llama-3.1-8B-Instruct, and Llama-3.1-70B-Instruct on the free-answer subset of the benchmark evaluated without the ramifications constraints using the zero-shot-CoT. The input is given in the formal language, i.e. PDDL. The results are categorized up by the action sequence length.

a 3.78% increase in performance, likely attributable to its initially low baseline performance. These findings suggest that the pretraining phase of LLMs, predominantly focused on natural language, plays a crucial role in shaping their reasoning capabilities. Consequently, formal language inputs, that deviate from this training paradigm, may hinder model performance.

## K.2 TEMPLATED LANGUAGE

| Action Seq. | GPT-4o | Llama-8B-Inst | Llama-70B-Inst |
|:---:|:---:|:---:|:---:|
| 1 | $76.92_{6.75}$ | $32.50_{7.40}$ | $63.41_{7.52}$ |
| 10 | $66.67_{7.55}$ | $19.51_{6.19}$ | $47.50_{7.89}$ |
| 19 | $64.10_{7.68}$ | $17.95_{6.15}$ | $25.64_{6.99}$ |

Table 18: Performance comparison of GPT-4o, Llama-3.1-8B-Instruct, and Llama-3.1-70B-Instruct on the free-answer subset of the benchmark evaluated without the ramifications constraints using the zero-shot-CoT. The input is given in the templatized language. The results are categorized up by the action sequence length.

Table 18 summarizes the performance of LLMs when inputs are presented in the templated formats described in Section 3.4 rather than in fully paraphrased natural language. The results indicate a consistent improvement across all models. Notably, GPT-4o achieves the highest gain, with an average performance improvement of 11.2%. Llama-3.1-8B-Instruct exhibits the second largest improvement, with a 6.5% increase, while Llama-3.1-70B-Instruct demonstrates a modest gain of 1.81%. These results suggest that templated inputs reduce the verbal reasoning burden on LLMs, leading to more accurate outputs.

