# OpenReview forum: "ActionReasoningBench: Reasoning about Actions with and without Ramification Constraints"
_ICLR.cc/2025/Conference — ICLR 2025 Poster_

### Official Review · Reviewer_24Bn · 2024-10-30

**Soundness:** 3
**Presentation:** 3
**Contribution:** 3
**Rating:** 8
**Confidence:** 3

**Summary:**

This paper introduces a new benchmark, ActionReasoningBench, to evaluate LLMs ability to reason about actions and change. This benchmark covers various RAC dimensions, including fluent tracking, state tracking, action executability, effects of actions, numerical RAC, and composite questions. Additionally, the authors introduce a new ramifications constrains to the reasoning task. The paper presents the evaluation of several sota LLMs across the proposed RAC dimensions proving that there is a significant margin of improvement for advanced LLMs such as GPT-4o and o1-preview.

**Strengths:**

- The paper is well written and easy to follow.
- It provides a comprehensive evaluation framework for LLMs' RAC capabilities, covering a wide range of tasks and difficulties. This benchmark allows for a thorough assessment of LLM performance and identifies areas for improvement such as new key challenges and LLMs limitations.

**Weaknesses:**

- I am not sure about the exclusive use of LLM to evaluate the free-form answers. It may introduce inaccuracies due to the limitations of language models. While the correlation between Llama-3.1-70B-Instruct and human evaluations is reported, it would be of interest to report evaluation from automatic metrics based on semantic similarity such as Bertscore (among others).
- The fine-grained methodology for evaluating RAC could be further enriched considering aspects such as causal reasoning (this is partially covered by “effects of actions”) and temporal reasoning (for example reasoning about concurrent actions).
- A short discussion beyond the examples provided in the appendix on how/why LLMs fail (with error analysis) the task would help better motivate the study beyond the performance in term of accuracy.

**Questions:**

Have you considered incorporating a combination of automatic evaluation metrics (e.g., BertScore) alongside the current evaluation?

---

> ### Author Response · Authors · 2024-11-24
>
> ## Weakness
> > I am not sure about the exclusive use of LLM to evaluate the free-form answers. It may introduce inaccuracies due to the limitations of language models. While the correlation between Llama-3.1-70B-Instruct and human evaluations is reported, it would be of interest to report evaluation from automatic metrics based on semantic similarity such as Bertscore (among others).
>
> Thank you for your thoughtful feedback. We have added F1-BERTScore along with LLM evaluation. Our decision to use Llama-3.1-70B-Instruct for free-form answer evaluation despite knowing it isn’t perfect is based on recent findings, which show that LLM-based evaluation aligns better with human judgments than traditional metrics like BLEU or BERTScore. These metrics often struggle with nuanced semantic understanding in complex reasoning tasks [1, 2, 3]. However, we believe it is important to see how these metrics perform in evaluating free-answers.
>
> Therefore, we have added F1-BERTScore results in Appendix F and included some examples where BERTScore fails to capture correct answers.
>
> We appreciate your input and hope this addresses your concerns.
>
> [1] Zheng, Lianmin, et al. "Judging llm-as-a-judge with mt-bench and chatbot arena." Advances in Neural Information Processing Systems 36 (2023): 46595-46623.
> [2] Liu, Yang, et al. "G-Eval: NLG Evaluation using Gpt-4 with Better Human Alignment." Proceedings of the 2023 Conference on Empirical Methods in Natural Language Processing. 2023.
> [3] Dubois, Yann, et al. "Length-controlled alpacaeval: A simple way to debias automatic evaluators." arXiv preprint arXiv:2404.04475 (2024).
>
> > The fine-grained methodology for evaluating RAC could be further enriched considering aspects such as causal reasoning (this is partially covered by “effects of actions”) and temporal reasoning (for example reasoning about concurrent actions).
>
> Thank you for your suggestion. We agree that there is a lot more that can be done to evaluate RAC. Ours is a key initial step.
>
> As stated in the limitations section of our paper, we acknowledge that there is potential to create even more sophisticated RAC questions by incorporating multiple reasoning types promising avenues for future exploration. However, our current focus is on establishing a solid foundation for RAC by addressing its fundamental aspects.
>
> > A short discussion beyond the examples provided in the appendix on how/why LLMs fail (with error analysis) the task would help better motivate the study beyond the performance in term of accuracy.
>
> Thank you for your feedback. We added to the preliminary error analysis presented in Section 5.1 on the ramification subset. We observed that GPT-4o struggles to generalize to unseen fluents, likely because these fluents are not part of the knowledge encoded during pre-training. This highlights the model’s reliance on pre-learned knowledge rather than robust reasoning over new information. o1-preview on the other hand, successfully reasons in most cases, figuring out the ramification constraints but often fails to mention negative fluents, i.e. fluents that are false in a state, even when explicitly mentioned to do so. In cases where it does make an error, it is always seen in a ramification fluent, indicating the struggles in reasoning over indirect effects of actions.
>
> We have added details regarding error analysis on the reasoning chains for non-ramification questions in Section 5. We found that LLMs excel in keeping track of the changes that happen in the state via an action. However, LLMs fail to identify whether an action is executable in the current state or not. Interestingly, the model correctly lists down the preconditions for the action but still fails to recognize that the action is inexecutable. We also observed that complex statements that involve multiple actions pose significant challenges. For eg, in the Grippers domain, consider the statement in the question - “Robot1 picks up the ball2 with lgripper2 and moves to room3, where it drops the ball2”. GPT-4o correctly identified the location of ball2 but failed to reason that lgripper2 becomes empty. While the final answer to this question was correct, this flaw reveals a potential overestimation of the model’s reasoning ability, which we now discuss explicitly.
>
> We have added more examples in Appendix J to illustrate these points, showing how errors often occur in intermediate reasoning steps, even when final answers appear correct.

---

> ### Author Response · Authors · 2024-11-24
>
> ## Questions
> > Have you considered incorporating a combination of automatic evaluation metrics (e.g., BertScore) alongside the current evaluation?
>
> We appreciate the reviewer’s suggestion. We have added F1-BERTScores in Appendix F for comparison. While traditional metrics like F1-BERTScore provide additional insights, our manual analysis reveals cases where they give high scores to incorrect responses and low scores to correct ones, making thresholding difficult. A combination of LLM-based evaluation and automatic methods (like BERTScore) could yield a better metric, but that is out of the scope of this paper. (Refer to Weakness 1 for details)

---

> > ### Comment · Reviewer_24Bn · 2024-11-25
> >
> > Thank you for your rebuttal. Most weaknesses are addressed in my opinion and I understand that this is an initial step towards more fine grained evaluation. I will raise my score.

---

### Official Review · Reviewer_QWAg · 2024-11-04

**Soundness:** 3
**Presentation:** 2
**Contribution:** 2
**Rating:** 5
**Confidence:** 3

**Summary:**

This paper introduces ActionReasoningBench, a benchmark for measuring LM reasoning on action sequences across multiple categories. The benchmark is created by adapting 8 planning domains from the International Planning Competition, and translating them into natural language using a combination of templates and language models. The natural language descriptions and questions are reviewed by annotators. The authors evaluate 4 models on these datasets: GPT-4o, Llama-8b-instruct, LLama-70B-instruct, and fine-tuned Llama-8b. They find that performance generally decreases as a function of action sequence length, and state-of-the-art LMs cannot solve the benchmark, performing particularly poorly on tasks that require numerical reasoning or a combination of other types of action-sequence reasoning (e.g. fluent tracking and action executability). Even GPT4 o1-preview struggles to perform well on inferring ramifications of actions.

**Strengths:**

A strength of this paper is in its comprehensiveness and quantity of analysis. The benchmark  explores many different aspects and ways of querying for action reasoning capabilities, and discovers many interesting trends. The dataset is very large and is able to identify reasoning capabilities which even the most state-of-the-art LLMs do not possess. The method for creating the dataset seems sound as there is a programmatic solution to these types of questions, and humans are used to validate paraphrases. The paper is overall clearly written.

**Weaknesses:**

1. The choices of tasks in the paper could be better motivated -- why are the 8 tasks present a representative survey of the important types of action reasoning? Why do we care about these 8 specifically? What are the consequences of LMs being able (or not able) to perform some of these types of reasoning? Why is numerical reasoning included (which is not fully related to action sequences), but not other types of reasoning (e.g. logical)?
2. Without a larger takeaway, this paper risks reading like simply a collection of empirical facts. This paper presents 7-8 separate takeaways in section 5, without too deeply investigating any single one. For example, what kind of errors do LLMs make on ramification fluents? What kind of intuition can we gain about LLMs and their limitations in reasoning from these errors. Having a more coherent high-level claim about all the takeaways from these results can be enlightening. Also having a table summarizing the limitations of LLMs in these 8 aspects could be useful.

**Questions:**

1. The use of Llama to generate question translations (into natural language) may affect the reasoning capabilities of LLMs, in particular requiring LLMs to use verbal reasoning skills to comprehend the questions. Are LLMs better at handling questions in natural language compared to templated?

---

> ### Author Response · Authors · 2024-11-24
>
> ## Weakness
> > The choices of tasks in the paper could be better motivated -- why are the 8 tasks present a representative survey of the important types of action reasoning? Why do we care about these 8 specifically? What are the consequences of LMs being able (or not able) to perform some of these types of reasoning?
>
> We appreciate the reviewer’s question about the motivation behind our selection of tasks and domains for evaluating reasoning about actions and change (RAC). Below, we provide a detailed explanation to clarify our choices.
>
> **8 Domains**
>
> The selection of the 8 domains represents varying levels of complexity as encountered by traditional AI systems, as shown in Appendix G. As mentioned in Section 3.3, these domains were sourced from the International Planning Competition and enable testing on both short and long sequences of actions, ensuring robust coverage of real-world scenarios.
>
> **6 Dimensions/Tasks**
>
> The 6 dimensions in RAC were chosen to comprehensively assess LLMs’ abilities to reason about actions. They include four fundamental dimensions (Fluent Tracking, State Tracking, Action Executability, and Effects of Actions) and two more advanced dimensions (Numerical RAC and Composite Questions) to pinpoint where LLMs struggle. These tasks are critical given the increasing use of LLMs in agentic systems, where reasoning over actions and their ramifications is essential. The tasks allow us to move beyond broad statements like "LLMs struggle with RAC" and instead identify precise dimensions where improvement is needed. Failures in specific dimensions, such as Action Executability or Numerical RAC, have direct implications for deploying LLMs in real-world, safety-critical, or decision-making applications.
>
> We hope this clarifies our rationale. Thank you for the opportunity to address this.
>
> > Why is numerical reasoning included (which is not fully related to action sequences), but not other types of reasoning (e.g. logical)?
>
> Thank you for your question. The inclusion of numerical reasoning in our benchmark is intended to showcase how our work can integrate reasoning capabilities to create more complex and diverse questions. Numerical reasoning is an example of extending the fundamental reasoning about actions and change (RAC) with additional dimensions, thus pushing the boundaries of what RAC tasks can evaluate. Logical reasoning on the other hand is implicitly covered in our current categories. For example, conjunction is covered when multiple properties need to be satisfied for an action to be executable. Negation and disjunction are covered in that in the absence of any of its preconditions an action is not executable.

---

> ### Author Response · Authors · 2024-11-24
>
> > Without a larger takeaway, this paper risks reading like simply a collection of empirical facts. This paper presents 7-8 separate takeaways in section 5, without too deeply investigating any single one. For example, what kind of errors do LLMs make on ramification fluents? What kind of intuition can we gain about LLMs and their limitations in reasoning from these errors. Having a more coherent high-level claim about all the takeaways from these results can be enlightening. Also having a table summarizing the limitations of LLMs in these 8 aspects could be useful.
>
> We thank the reviewer for their valuable feedback. Below, we address the concerns raised in detail:
>
> 1. **Larger Takeaway**: The larger takeaway from our work is that RAC is a key aspect of AI (since the beginning of AI) and is not an NP-hard or NP-complete (or beyond) problem, such as planning, and plays an important role in commonsense reasoning and planning, yet LLMs still struggle with RAC. Thus it should be among the reasoning benchmarks of LLMs.
>    From our results, we found that while LLMs can perform reasonably well on direct or simpler reasoning tasks, their ability to handle indirect effects and complex dependencies remains limited, revealing critical gaps in their reasoning capabilities.
>
> 2. **Error Analysis on Ramificiatinon Constraints**: We share insights from the manual analysis on the ramification subset of the benchmark in Section 5.1. Findings include:
>
>     2.1. GPT-4o fails to reason over the new ramification fluents, likely because these fluents were unseen during pre-training. Since the domains were sourced from IPC, it is very likely that they have been seen by GPT-4o during its pretraining. This indicates a reliance on pre-trained knowledge for reasoning and an inability to generalize to novel fluents.
>
>     2.2. o1-preview demonstrates better reasoning abilities but struggles with negative fluents, i.e. fluents that are false in a state, even when explicitly asked in the question. We observe correct responses when the question only involves positive fluents, but we still observe some incorrect reasoning steps, indicating future scope.
>
> 3. **Error Analysis on Non-Ramification Constraints**: We have added error analysis details in Section 5. We found that LLMs excel at tracking state changes via actions but struggle to determine action executability, despite correctly identifying preconditions. This suggests that LLMs don’t inherently understand RAC but can extrapolate the effects of actions from the domain description. Moreover, multi-action statements, which are more natural sounding, highlight the struggles faced by LLMs as they try to determine the final effects of multiple actions in one go.
>
> 4. **Limitation across every task**: Upon manually analyzing the reasoning chains of every task in our benchmark, we found that LLMs do reasonably well in keeping track of the state even after a long sequence of actions is performed. This results in a high accuracy in Fluent Tracking, State Tracking, and Effects of Actions. However, they fail to determine whether an action is executable or not. Interestingly, LLMs correctly list down the preconditions for these actions but fail to match them against the description provided in the prompt. For Numerical RAC, it has been shown in previous works that LLMs struggle in counting tasks, potentially due to the bias of numbers in the multiple of 5 or 10 (10, 15, 20, 25, etc.) that are seen on the internet [3]. For Composite Questions, we observe that the questions involving Action Executability were more difficult to answer as the LLM was unable to identify the correct inexecutable action resulting in the wrong answer.
>
> We have updated Section 5 to reflect these changes as well.
>
> [3] McCoy, R. Thomas, et al. "Embers of autoregression: Understanding large language models through the problem they are trained to solve." arXiv preprint arXiv:2309.13638 (2023).

---

> ### Author Response · Authors · 2024-11-24
>
> ## Questions
> > The use of Llama to generate question translations (into natural language) may affect the reasoning capabilities of LLMs, in particular requiring LLMs to use verbal reasoning skills to comprehend the questions. Are LLMs better at handling questions in natural language compared to templated?
>
> Thank you for the question. The motivation behind converting the templated questions into natural language is to introduce more diversity to the questions. Templated questions can easily be parsed with a Python wrapper or even by finetuning a small LLM. To prevent this and verify the actual RAC capabilities of these LLMs, we paraphrased them using Llama-3.1-70B-Instruct. Moreover, this transition improves readability by reducing redundancy (e.g., “The truck t1 goes from loc1 to loc2. The truck t2 goes from loc1 to loc2” becomes “Both trucks t1 and t2 go from loc1 to loc2”). We verified this through a manual review of 65 samples and subsequent evaluation by annotators for naturalness. Our observations indicate that this paraphrasing maintains the essence of the original question while improving their fluency.
>
> To measure the impact of additional verbal reasoning skills that are introduced by this translation, we conducted an experiment using 10% of the data across all question categories and action sequence lengths with the original templated data and have added in Appendix K.2. Interestingly, the biggest LLM, GPT-4o showed the highest improvement (~11%), and the smaller open-source LLMs show a smaller increase–6% and 1% respectively for Llama-3.1-8B-Instruct and Llama-3.1-70B-Instruct. We believe that while the paraphrasing slightly increases linguistic complexity, it better represents the types of questions LLMs face in real-world applications, aligning with the broader goals of this benchmark.

---

### Official Review · Reviewer_sFGq · 2024-11-04

**Soundness:** 3
**Presentation:** 3
**Contribution:** 3
**Rating:** 6
**Confidence:** 3

**Summary:**

This paper proposes a reasoning benchmark for reasoning about actions and changes. The focus of this benchmark is specifically on tasks that require state tracking and commonsense, which often requires a full understanding of the current state. The paper aggregates this benchmark using the IPC advance planning competition challenge, by using a pipeline to convert them to natural language and testing it on the latest LLMs. The benchmark reveals headroom for LLMs in the context of state tracking, specifically when ramification constraints are incorporated in the challenge.

**Strengths:**

- The paper is a step forward towards producing a more challenging benchmark in the domain of reasoning about actions and changes, compared to the previous PlanBench and TRAC work.

- It also incorporates additional metrics in state tracking that is not part of previous benchmarks.

- The latest models are also thoroughly evaluated with a good sense of headroom, and will provide a meaningful resource for making progress in this domain.

**Weaknesses:**

- Related work is poorly written. It doesn’t cover the nature of the dataset in the context of existing benchmarks (I had to read the other papers to get a good context). It was hard to identify why this benchmark was the right direction to continue as a benchmark. I highly recommend the authors to expand the related work section placing the contribution in context (particularly for readers like me, who don’t have the full context).

- Regarding the benchmark itself, the paper doesn’t give enough details on how challenging the individual domains are. It is hard to judge the contribution of the paper well, without those details. Specifically, how were these tasks solved before LLMs were used for these domains? What progress have they made ? While LLMs can be a general model for these tasks, how would they different from building custom models for these tasks ?

Note: I still very much like this paper, and happy to increase the score if these points above are addressed well.

**Questions:**

- Why should an LLM alone be used to solve this and not in combination with other systems ? What aspects of the challenges in these individual domains are LLMs most suitable ? Could you please elaborate a bit on this ?

- What are the current state-of-art (including non-LLM) methods for this ? How do LLMs perform in comparison to them ?

- Are there any Insights into why numerical RAC performance doesn't seem to improve even after fine-tuning? I was wondering if it was because “number of possible states” is a well-posed problem ? Love to know the authors’ opinion on this

- (Line 193) What are negative fluents ?

- What happens if we don’t convert the examples into natural language ?  Does the LLM perform any well at all in this task without the natural language conversion step ?

---

> ### Author Response · Authors · 2024-11-24
>
> ## Weakness
> > Related work is poorly written. It doesn’t cover the nature of the dataset in the context of existing benchmarks (I had to read the other papers to get a good context). It was hard to identify why this benchmark was the right direction to continue as a benchmark. I highly recommend the authors to expand the related work section placing the contribution in context (particularly for readers like me, who don’t have the full context).
>
> Thank you for the valuable feedback. We have expanded the Related Works to position our benchmark in relation to existing benchmarks. We now emphasize the unique aspects of our benchmark, including its broader domain coverage, longer action sequences, and novel dimensions like Numerical Reasoning and Ramification Constraints.
>
> Additionally, Table 1 has been referenced explicitly to highlight how our work advances prior efforts by systematically addressing gaps in evaluating LLMs for RAC tasks. We believe these revisions will make the contribution and context more accessible to all readers.
>
> > Regarding the benchmark itself, the paper doesn’t give enough details on how challenging the individual domains are. It is hard to judge the contribution of the paper well, without those details. Specifically, how were these tasks solved before LLMs were used for these domains? What progress have they made? While LLMs can be a general model for these tasks, how would they be different from building custom models for these tasks?
>
> We thank the reviewer for raising these insightful questions. Below, we provide clarifications and additional discussions to address the concerns, which we incorporated in the updated Section 5 of our manuscript:
>
> 1. **Details on Domain Complexity**:
>    The paper includes a description of State Complexity in Appendix G, where we quantify the complexity of each domain based on the number of fluents, actions, and objects. This metric reflects how challenging the domain is for traditional AI systems. Moreover, Section 5 elaborates on how LLMs perform across domains, showing that their performance is not strictly correlated with State Complexity. For example, harder domains (high State Complexity) like Grippers yield better performance, highlighting different reasoning paradigms for LLMs and traditional systems.
>
> 2. **Previous Solutions Without LLMs**:
>    In the past humans were involved in converting natural language descriptions to a formal representation and logic. Logical representations and reasoning rules were then identified from the literature and their corresponding implementation is then used to solve them. Some examples of these logical reasoning systems are Answer Set Solver and SAT solver. If there are no logical representations or reasoning rules present, then humans have to develop them. These systems are accurate but can’t process natural language, limiting their practical application.
>
> Traditional AI systems—those based on such formal languages, like Planning Domain Definition Language (PDDL)—can then solve these tasks accurately. We leverage this fact to create our benchmark synthetically with the assurance that the ground truth is correct. However, such systems cannot interpret and respond in natural language, which limits their practical applications. A critical advantage of LLMs is their capability to process NL, enabling general-purpose use.
>
> 3. **Custom Models**:
>     Custom models can be implemented via finetuning or using external tools like PDDL solvers to critique the LLMs. But these methods don’t give insights into the LLM’s performance and more importantly what LLMs struggle with within RAC. We believe that prompting LLMs with different techniques (zero-shot-CoT and few-shot-3, in our case) can help highlight areas where LLMs struggle which can lead to more smarter and more powerful LLMs.
>
> As shown in our original submission (in Tables 3 and 4) (now Tables 3 and 5) significant improvement is observed in the finetuned Llama-3.1-8B in our experiments compared to GPT-4o. Thus, custom models outperform basic LLMs. However, the goal of our paper is to evaluate how well LLMs do in reasoning about actions. Through this paper, we hope that LLM developers will use our dataset as one of their reasoning benchmarks.

---

> ### Author Response · Authors · 2024-11-24
>
> ## Questions
> > Why should an LLM alone be used to solve this and not in combination with other systems? What aspects of the challenges in these individual domains are LLMs most suitable? Could you please elaborate a bit on this?
>
> Thank you for this question.
>
> As LLMs are being used in a variety of applications, they should be able to do  RAC as that is a basic step of many commonsense reasoning scenarios (as noted in few seminal books [1][2]) and more complex reasoning such as planning. The decision to evaluate LLMs independently intentionally highlighted their standalone capabilities in RAC without relying on external systems. While integrating LLMs with formal languages like PDDL could enhance performance, such approaches shift the focus from evaluating LLMs' own reasoning abilities.
>
> Our findings indicate that LLMs perform well in Fluent Tracking, State Tracking, and Effects of Actions, demonstrating their strength in keeping track of changes. However, they struggle with Action Executability, Numerical RAC, and Composite Questions. This limitation is even more pronounced under ramification constraints, where indirect dependencies challenge the models further.
>
> We have elaborated on these findings in Section 5, analyzing trends across question categories and highlighting where LLMs excel and fail in RAC tasks.
>
> [1] Mueller, Erik T. Commonsense reasoning: an event calculus based approach. Morgan Kaufmann, 2014.
> [2] LReiter, Raymond. Knowledge in action: logical foundations for specifying and implementing dynamical systems. MIT press, 2001.
>
> > What are the current state-of-art (including non-LLM) methods for this? How do LLMs perform in comparison to them?
>
> Non-LLM methods, such as ASP or PDDL-based solvers, are capable of solving the benchmark fully accurately when provided with formal representations, as the dataset was created using these systems. However, these approaches require input in formal languages, limiting their usability for natural language tasks.
>
> In comparison, LLMs operate directly on natural language input, offering greater usability and adaptability. Despite this, they perform less effectively on complex tasks, such as reasoning with ramifications (e.g., o1-preview achieves only 18.4% accuracy on such questions). While traditional solvers excel in precision, LLMs provide a broader, more flexible framework, though improvements are needed to close the performance gap in complex reasoning.
>
> > Are there any Insights into why numerical RAC performance doesn't seem to improve even after fine-tuning? I was wondering if it was because “number of possible states” is a well-posed problem? Love to know the authors’ opinion on this
>
> Thank you for the question. Yes, calculating the number of states/actions is a well-posed problem. However, while LLMs excel at simpler well-posed problems, their performance degrades with longer sequences of actions, and in questions requiring numerical precision. The task of “counting” which is integral to Numerical RAC, has proven particularly challenging for LLMs, as shown in [3].
>
> Our results show that finetuning improves Action Executability–indicating better identification of executable or inexecutable actions–but when asked to count the number of executable/inexecutable actions, we see marginal improvements. This highlights the inherent challenges LLMs face in numerical reasoning, even after fine-tuning.
>
> [3] McCoy, R. Thomas, et al. "Embers of autoregression: Understanding large language models through the problem they are trained to solve." arXiv preprint arXiv:2309.13638 (2023).
>
> > (Line 193) What are negative fluents?
>
> Thank you for the question. Negative fluents refer to fluents that are false. For example, in the Blocksworld domain, if block b2 is placed on block b1, the fluent “clear(b1)” becomes false, leading to a negative fluent: “b1 is **not** clear.”
>
> In the 205 (originally 193) line, we meant that we also generate questions about negative fluents (false fluents), in addition to questions about all fluents (both true and false). We hope this clears up the confusion.
>
> > What happens if we don’t convert the examples into natural language? Does the LLM perform any well at all in this task without the natural language conversion step?
>
> Thank you for your question. We experimented on 10% of the data across all question categories and action sequence lengths, using formal representations (PDDL) as input instead of natural language. This resulted in a performance drop of approximately 7%. This decline is likely because LLMs are predominantly trained in natural language, making their reasoning stronger in that format. These results highlight the importance of natural language conversion for leveraging LLMs' reasoning capabilities effectively. We have added this study in Appendix K.

---

> > ### Comment · Reviewer_sFGq · 2024-11-27
> >
> > The authors have addressed my concerns sufficiently. I am raising my score.

---

### Official Review · Reviewer_byEu · 2024-11-04

**Soundness:** 3
**Presentation:** 2
**Contribution:** 4
**Rating:** 8
**Confidence:** 4

**Summary:**

This paper introduces ActionReasoningBench, which evaluates LLMs' ability to reason about actions and change in PDDL-based domains. The benchmark introduces novel ramification fluents which require deeper reasoning to infer correctly, such as fluents which are dependent on combinations of other properties, and other properties like "if object is at location a, it cannot also be at location b". The results show that LLMs predict the standard fluents well, but are almost completely failing at such ramification fluents.

**Strengths:**

- The paper studies an important but overlooked problem of evaluating RAC capabilities of LLMs in a systematic way, beyond measuring surface RAC. The paper demonstrates through the difference in performance on basic fluents and the ramification fluents, that this is indeed a weakness of current LLMs.

- The paper's conceptualization and discussion of ramification fluents is useful, and the benchmark has potential to lead to interesting new research such as entirely new models for RAC.

**Weaknesses:**

I have two main concerns with the paper in its current form:

- Organization: The organization of the paper could be improved, as the main results of the paper appears to be Table 5 (the ramification results), but these experiments are conducted on a smaller set of models and limited to free-answer. The experiments seem a bit incomplete. Other sections like "Performance across Fluent Categories" (l415) should be shown as table/figure.
- Human baseline: It's difficult to gauge the human upper bound on the benchmark with ramifications, and I would suggest that a human baseline would offer a better way to contextualize the low numbers shown by LLMs.

**Questions:**

- In Table 3, are only the base fluents inclulded in the "Fluent tracking" category?
- Is my understanding correct that when you refer to "introducing ramification constraints", it indicates introducing the ramification fluents in the benchmark?

---

> ### Author Response · Authors · 2024-11-24
>
> ## Weakness
> > Organization: The organization of the paper could be improved, as the main results of the paper appears to be Table 5 (the ramification results), but these experiments are conducted on a smaller set of models and limited to free-answer. The experiments seem a bit incomplete.
>
> Thank you for your feedback. We have improved the paper and have added additional experiments on the binary subset containing the ramification questions. We would like to explain our design choices below:
>
> 1. **Focus on RAC (Tables 3 and 4) (Now 3 and 5):** We want to highlight the challenges faced by LLMs with even straightforward RAC problems, emphasizing the need for a more comprehensive benchmark like ours.
>
> 2. **Ramification Results (Table 5) (Now Table 4):** We focused on ramifications as an extension of the benchmark. For these experiments, we restricted evaluations to GPT-4o (the best-performing model on non-ramification tasks) and o1-preview (a model optimized for reasoning). The free-answer evaluation was chosen as it better captures the challenges introduced by ramifications compared to binary tasks.
>
> 3. **Additional Results:** We have added binary subset results in Table 4 to complement the free-answer evaluations, addressing your concern about limited scope.
>
> We hope this clarifies our approach and highlights the progression from evaluating simple RAC to introducing the novel challenges of ramifications. Thank you for your valuable input.
>
> > Other sections like "Performance across Fluent Categories" (l415) should be shown as table/figure.
>
> Thank you for your suggestion. We have revised the paper to add a figure (Fig 2) to highlight these results.
>
> > Human baseline: It's difficult to gauge the human upper bound on the benchmark with ramifications, and I would suggest that a human baseline would offer a better way to contextualize the low numbers shown by LLMs.
>
> Thank you for the suggestion. During the rebuttal period, we conducted a human performance study on a randomly sampled subset of our benchmark, including all question categories and action sequence lengths for free-answer questions. Three independent annotators participated in this study, achieving an average accuracy of ~79%, representing a 21% improvement over GPT-4o's performance in the non-ramification subset.
>
> A closer analysis of the human responses revealed consistent performance across ramification and non-ramification subsets and different action sequence lengths. However, errors were more common in cases where the answer state space was large, even though annotators accurately represented the problem states in their diagrams while solving the problem. Additionally, numerical RAC questions posed challenges where the answers were in the hundreds. Interestingly, when annotators received feedback on their responses, they were able to correct most of their mistakes, raising their accuracy from 79% to near-perfect levels. This adaptability highlights a significant difference between human reasoning and LLM capabilities, as LLMs do not exhibit this ability to refine their understanding with feedback, further underscoring the superior grasp of RAC concepts by humans.
> ## Questions
> > In Table 3, are only the base fluents included in the "Fluent tracking" category?
>
> The "Fluent Tracking" category in Table 3 includes all fluents: Static Properties, Base Fluents, Derived Fluents, Self-Derived Fluents, and Mixed Fluents. It is not limited to base fluents.
>
> > Is my understanding correct that when you refer to "introducing ramification constraints", it indicates introducing the ramification fluents in the benchmark?
>
> By "introducing ramification fluents," we refer to both defining new fluents and adapting existing ones to reflect indirect dependencies:
> 1. **New Fluents**: For example, in Blocksworld, we introduce a new fluent, "stable", to represent a block being stable when “clear” and “on the table”.
> 2. **Converted Existing Fluents**: Fluents like "clear" are transformed into derived fluents, determined by dependencies (e.g., a block is “clear” if “nothing is on it” and it’s “not held”) rather than direct action effects.
>
> In the ramification subset, these fluents are not directly modified by actions but are derived through other fluents, capturing indirect effects. This approach emphasizes the complexity of reasoning with ramifications, aligning with our benchmark's goals.
>
> We hope this clarifies our approach. Thank you for raising this point.

---

> > ### Comment · Reviewer_byEu · 2024-11-27
> >
> > I appreciate the authors' response and clarifications, and they have addressed my concerns/questions.

---

### Meta-Review · Area_Chair_4dTc · 2024-12-19

**Metareview:**

The paper presents a new benchmark that challenges LLMs by asking them to reason about environments where there is grounded actions that change the world - especially scenarios involving counting and composing various questions. The benchmark is mostly synthetically labelled via an LLM and while their argument that human evaluation is expensive - a small correlational study between human annotators and LLM outputs would have been useful. Most of the reviewers concerns were addressed during the rebuttal period and I'd encourage the authors to incorporate the changes they mention - especially the motivation stemming from the most closely related work.

**Additional Comments On Reviewer Discussion:**

Reviewers 24Bn and sFGq raised their scores after the reviewer discussion and reviewer QWAg who recommended rejection did not engage with the rebuttal. I am of the overall opinion that most of the reviewers clarifications were sufficiently addressed by the authors during the discussion period.

---

### Decision · Program_Chairs · 2025-01-22

Accept (Poster)